# A DAP5/eIF3d alternate mRNA translation mechanism promotes differentiation and immune suppression by human regulatory T cells

Viviana Volta [1,7], Sandra Pérez-Baos [2,7], Columba de la Parra[3], Olga Katsara[2], Amanda Ernlund[4], Sophie Dornbaum[2] & Robert J. Schneider [2,5,6 ✉]

Regulatory T cells (Treg cells) inhibit effector T cells and maintain immune system homeostasis. Treg cell maturation in peripheral sites requires inhibition of protein kinase mTORC1 and TGF-beta-1 (TGF-beta). While Treg cell maturation requires protein synthesis, mTORC1 inhibition downregulates it, leaving unanswered how Treg cells achieve essential mRNA translation for development and immune suppression activity. Using human CD4+ T cells differentiated in culture and genome-wide transcription and translation profiling, here we report that TGF-beta transcriptionally reprograms naive T cells to express Treg cell differentiation and immune suppression mRNAs, while mTORC1 inhibition impairs translation of T cell mRNAs but not those induced by TGF-beta. Rather than canonical mTORC1/eIF4E/ eIF4G translation, Treg cell mRNAs utilize the eIF4G homolog DAP5 and initiation factor eIF3d in a non-canonical translation mechanism that requires cap-dependent binding by eIF3d directed by Treg cell mRNA 5′ noncoding regions. Silencing DAP5 in isolated human naive CD4+ T cells impairs their differentiation into Treg cells. Treg cell differentiation is mediated by mTORC1 downregulation and TGF-beta transcriptional reprogramming that establishes a DAP5/eIF3d-selective mechanism of mRNA translation.

[1] Synthis LLC, 430 East 29th Street, Launch Labs, Alexandria Center for Life Sciences, New York, NY 10016, USA. [2] Department of Microbiology, NYU Grossman School of Medicine, New York, NY 10016, USA. [3] Department of Chemistry, Herbert H. Lehman College, City University of New York, The Graduate Center, Biochemistry Ph.D. Program, City University of New York, New York, NY 10016, USA. [4] Johns Hopkins Applied Physics Lab, 11000 Johns Hopkins Road, Laurel, MD 20723, USA. [5] Perlmutter Cancer Center, NYU Grossman School of Medicine, New York, NY 10016, USA. [6] Colton Center for Autoimmunity, NYU Grossman School of Medicine, New York, NY 10016, USA. [7] These authors contributed equally: Viviana Volta, Sandra Pérez-Baos. ✉email: robert.schneider@nyumc.org

Treg cells constitute ~5% of CD4[+] T lymphocytes that exert antiproliferative action on activated immune cells. They act by contact-dependent and independent mechanisms, thereby maintaining immune system homeostasis, inhibiting effector T cells in the periphery, controlling excessive responses to foreign antigens, and preventing autoimmune disease[1,2].

Treg cells can be categorized into three subsets: thymus-derived Treg cells (tTreg cells) known as natural Treg cells, peripherally derived Treg cells (pTreg cells) and in vitro-induced Treg cells (iTreg cells)[3]. tTreg cells and pTreg cells occur naturally in animals and have distinct functions. tTreg cells are mainly involved in curtailing systemic autoimmunity, whereas pTreg cells suppress localized inflammatory responses[4,5]. Antigen activated naive CD4[+] T cells differentiate into effector T cells or Treg cells, controlled in part by local metabolic parameters and cytokines[6], although lineages demonstrate considerable phenotypic plasticity[7,8]. mTOR complex 1 (mTORC1) inhibition is required to generate and expand Treg cells, which occurs naturally by retinoic acid, short-chain fatty acids, metabolic stress, or by pharmacologic inhibitors that block mTORC1 activity[9–13].

mTOR is a protein kinase in the PI3K-Akt pathway that forms two complexes, mTORC1 and mTORC2[14]. mTORC1 is activated by growth factor signaling via the PI3K-AKT axis and MAPK pathways, and inhibited by stresses such as hypoxia, low energy status, and reduced nutrient levels. Activated mTORC1 promotes ribosome biogenesis, mRNA translation, DNA replication, and repair, and suppresses autophagy[15]. mTORC2 is activated by PI3K and regulates actin organization, cell motility, Akt activity, and other kinases[15]. Downregulation of mTORC1 activity impairs the transition of naive T cells to effector CD4[+] T cells, blocking the development of Th1, Th2, and Th17 cells, and instead skews naive CD4[+] cells and effector CD4[+] cells into pTreg cell reprogramming[16]. Numerous studies have shown that mTORC1 activity suppresses pTreg cell development, whereas mTORC1 inhibition promotes it[9,16–20]. However, mTORC1 inhibition also blocks cap-(m[7]G)-dependent mRNA translation, the major mechanism for protein synthesis, impairing protein synthesis just when pTreg cells need to differentiate and acquire immune-suppressing activity, which requires translation of mRNAs that specify these functions.

Little is known regarding the role of translational control in immune cell development and function. In fact, although only a fraction of the mammalian mRNAs actually participate in translation, particularly during cell stress and developmental cell fate decisions[21], characterization of the genome-wide translation signature, or translatome, is often overlooked. A number of specialized translation mechanisms have been described that allow for selective mRNA translation, but which remain largely unexplored in immune cell development, including internal ribosome entry site (IRES)-mediated mRNA translation and an alternate mechanism of cap-dependent but eIF4E/mTORC1-independent mRNA translation carried out by the DAP5/eIF3d complex[22].

mTORC1 stimulates translation initiation in part by phosphorylation (inactivation) of the 4E-BPs, negative regulators of cap-binding protein eIF4E[23]. Inhibition of mTORC1 results in hypo-phosphorylation (activation) of the 4E-BPs, which then bind and sequester eIF4E, preventing their interaction with the cap-dependent pre-initiation complex, thereby blocking recruitment of ribosomes to mRNAs[23]. The triggering of Treg cell development by mTORC1 inhibition, therefore, occurs at a time in which overall Treg cell protein synthesis is significantly inhibited, suggesting that there may be a mechanism for selective translation of Treg cell fate-determining mRNAs.

Here we show that in activated human CD4[+] naive T cells, mTORC1 downregulation of mRNA translation, in combination with TGF-beta transcriptional reprogramming, mediates the development of strongly immune-suppressive Treg cells. Comparative transcriptomic and translatomic studies demonstrate that Treg cell fate-determining mRNAs are translationally privileged, and can utilize an alternate mechanism of cap-dependent mRNA translation that is mTORC1/eIF4E-independent, directed by the DAP5/eIF3d complex, and essential for human iTreg-cell development from uncommitted CD4[+] T cells.

## Results

### Inhibition of mTORC1 but not mTORC1/2 induces CD4[+]FOXP3[+]CD25[+]CD127[−] cells from activated human T cells.

Functional human and mouse Treg cells can be generated in culture by the mTORC1 inhibitor rapamycin[24]. We confirmed that primary human lymphocytes can be differentiated to Treg cells in culture by inhibition of mTORC1 with inhibitor RAD001, but found that inhibition of both mTORC1 and 2 with the dual inhibitor PP242 blocks Treg cell development. Human naive CD4[+] T cells were isolated from peripheral blood mononuclear cells (PBMCs) obtained from healthy donors, activated by CD3 and CD28 co-stimulation in the presence of IL-2, and treated with escalating doses of RAD001 or PP242 for 4 days[25]. The phosphorylation status of mTORC1 downstream targets rpS6 and 4E-BP1, and mTORC2 target AKT, were assessed as a measure of mTOR activity (Fig. 1a). Untreated cells (effector, Th0 T cells) exhibited high phosphorylation levels of downstream mTORC1 targets 4E-BP1 and rpS6, consistent with activation of lymphocyte protein synthesis[26]. Optimal dosing with mTORC1 inhibitor RAD001 (20 nM) and mTORC1/2 inhibitor PP242 (1 μM) were chosen based on strong reduction in 4E-BP1 phosphorylation without reduced 4E-BP1 protein levels seen with very high levels of mTOR inhibitors, presumably by protein degradation[27]. AKT phosphorylation was increased by mTORC1 inhibition with RAD001, consistent with a reported activating feedback loop signaling to the insulin receptor[28]. The weak reduction in AKT P-S473 with dual mTORC1/2 inhibition is consistent with other reports which showed that lymphocytes have a unique response to mTOR inhibitors[29]. The levels of mTOR inhibitors identified for these studies are consistent with those employed in protein synthesis and clinical studies.

Human naive CD4[+] T cells isolated and activated as above, were treated with mTORC1 or mTORC1/2 inhibitors for 3 d, replated, expanded, and monitored for Treg cell markers FOXP3 and CD25 (Fig. 1b). Co-treatment with activated TGF-beta was required to induce a substantial increase in the number of FOXP3[+]CD25[+] on CD4[+] T cells to ~35% of the population, a requirement that distinguishes mouse and human Treg development[9,30,31]. The strong increase in FOXP3[+]CD25[+]CD4[+] T cells seen at d 6 decreased by d 13 (Fig. 1b) but not if re-stimulated every 7 d with αCD3/αCD28 (see later Fig. 2d), consistent with a requirement for continuous engagement of the TCR and mTOR inhibition in Treg cell differentiation[24]. All studies hereafter, therefore, used continuous activation with TGF-beta, which was essential to induce FOXP3[+]CD25[+] expression and mTORC1 inhibition, and every 7 d stimulation with αCD3/αCD28 to maintain high levels of activated Treg cells. Treg cells were purified by magnetic sorting following differentiation. The use of biomarker FOXP3 is warranted in this setting because it is considered a hallmark of Treg cell differentiation, although it is required but not sufficient to induce the Treg cell phenotype[8]. In human T cells, FOXP3 expression is also transiently induced during activation of CD4[+] T cells[32,33].

mTOR activity is crucial for eIF4E-directed mRNA translation. It was therefore unexpected that mTORC1 inhibition did not block T-cell proliferation, either alone or in combination with activated TGF-beta, which was reduced several-fold compared to untreated T cells (Fig. 1c). Dual mTORC1/2 inhibition fully

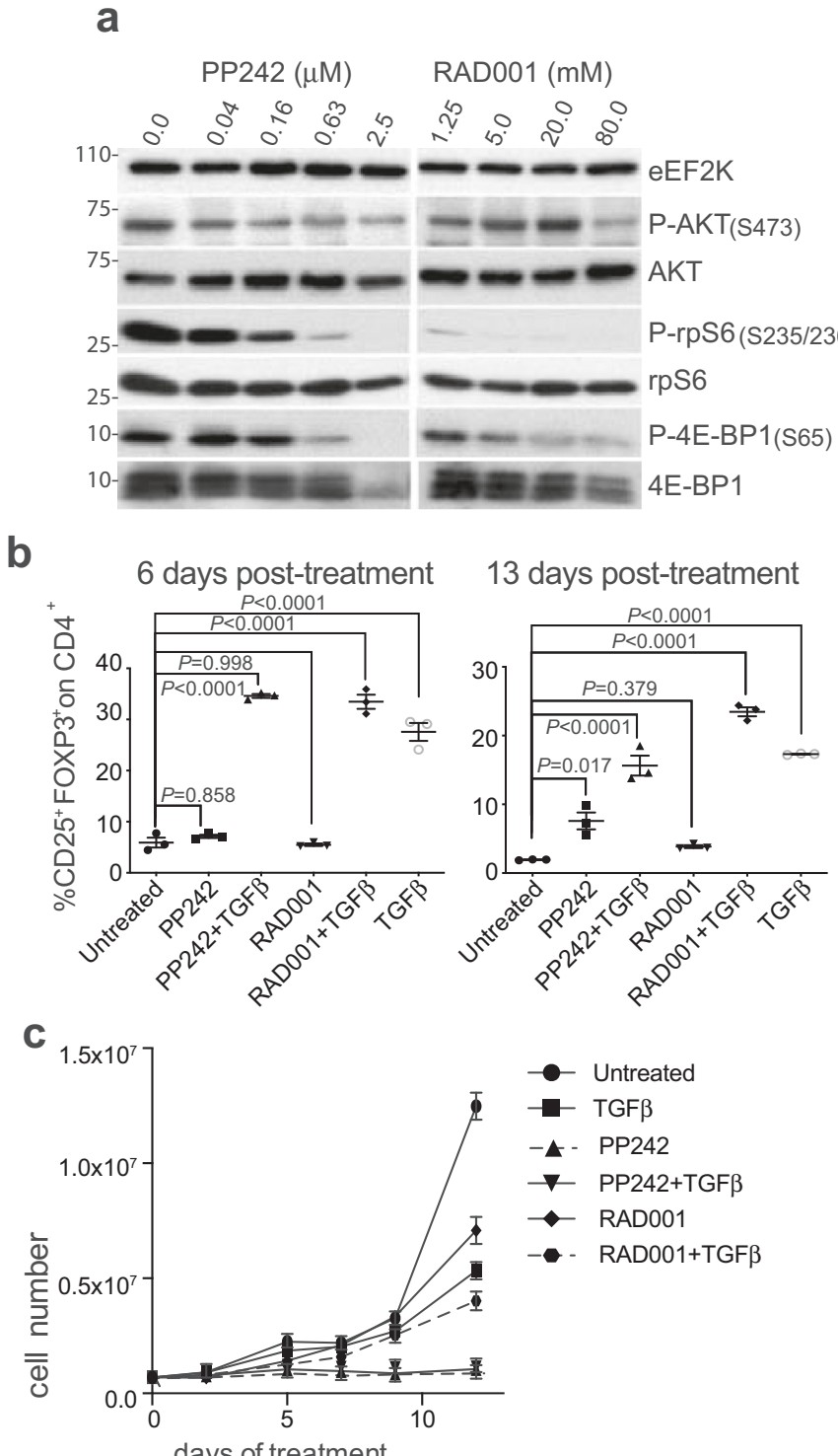

**Fig. 1 mTOR inhibition impairs human T cell differentiation. a** Titration of mTORC1 inhibitor RAD001 and dual mTORC1/2 inhibitor PP242 in human CD4[+] T cells. Human naive CD4[+] T cells were isolated from PBMCs of healthy donors as shown in Supplementary Fig. 1a. Isolated CD4[+] T cells were treated with plate-bound αCD3 antibody, soluble αCD28, and 150 U/ml IL-2, treated with increasing doses of the indicated drugs or DMSO, for 4 d and equal amounts of whole-cell lysates analyzed by immunoblot. A representative immunoblot from one donor is shown. Four independent donors were evaluated. **b** RAD001 mTORC1 inhibition +TGF-beta treatment (TGFβ in figures) increases CD4[+]FOXP3[+] T cells. Isolated CD4[+] T cells were activated and expanded for 13 d in the presence of DMSO, 1 μM PP242, 20 nM RAD001, 2 ng/ml TGF-beta or TGF-beta plus mTOR inhibitors with 150 U/ml IL-2 as needed. At 6 and 13 d post activation, cells were stained for CD4, CD25, FOXP3, and viability analyzed by flow cytometry. $P$ values were determined by statistical analysis using one-way ANOVA tests with Dunnett post-ANOVA test determination. Values shown are mean with standard error of the mean (SEM) of three independent studies. **c** mTORC2 is required for T-cell proliferation. CD4[+] T cells were activated, treated, and expanded as described in **b**. Shown is a time course of total cell number (cell concentration × culture volume). Data represent three independent studies with SEM shown. Source data are provided as a Source Data file.

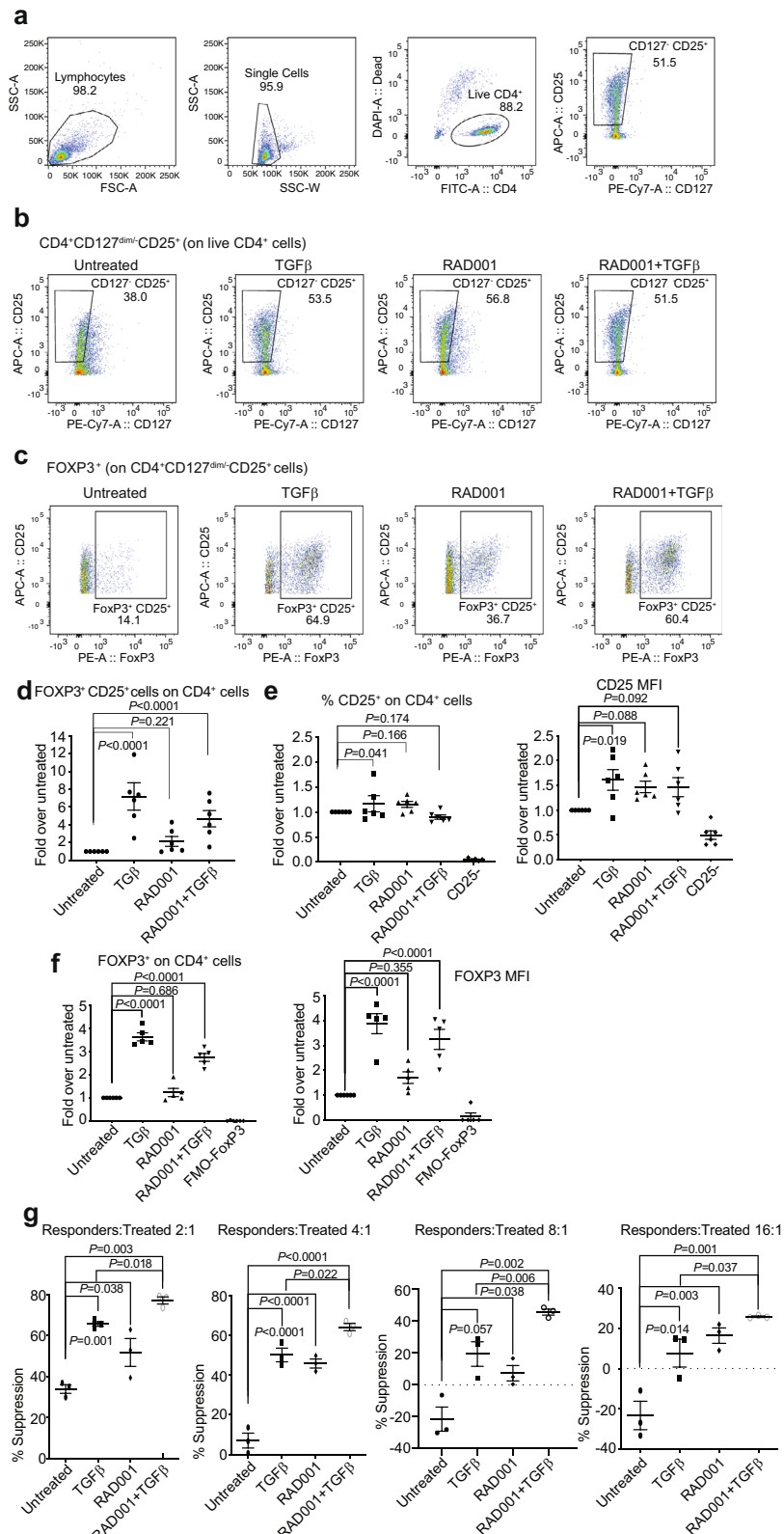

blocked T-cell proliferation, alone or in combination with TGF-beta, indicating that mTORC2 activity is required for lymphocyte proliferation. Dual mTORC1/2 inhibition was therefore not further studied in Treg cell differentiation.

**mTORC1 inhibition with TGF-beta exposure generates highly suppressive human Treg cells.** Although FOXP3 and CD25 are

widely used markers for both human and mouse Treg cells, in humans CD25 is also expressed in effector T cells, and FOXP3 is transiently expressed during activation of human lymphocytes[32,34]. We, therefore, included the absence of the CD127 marker to identify Treg cells because it is diagnostically downregulated in human Treg cells[35,36]. Naive CD4[+] cells from six different human donors were activated at d 1 and d 7, subjected to constant treatment with

**Fig. 2 mTORC1 inhibition plus TGF-beta treatment increases Treg-cell development and immune-suppression activity. a** Representative parental populations of CD24+CD127−/dimCD25+ cells (representative of three donors). Human naive CD4+ lymphocytes were activated at d 1 and d 7 by αCD3, soluble αCD28, and 150 U/ml IL-2 and cultured for 12 d. Treatments of 20 nM RAD001, 2 ng/ml activated TGF-beta, RAD001+TGF-beta or DMSO (untreated) were added during activation and subsequent expansions as needed. IL-2 was refreshed every 2–3 d. After 12 d, cells were harvested, treated cells enriched for Treg cells by magnetic-labeling recovery. Sorted cells were stained and analyzed by flow cytometry. Total lymphocytes were gated for doublet exclusion, live CD4+ T cells, and CD25+CD127− cells as shown. **b** TGF-beta or RAD001 treatments increase the percentage of CD25+CD127− in CD4+ population. Quantitation of CD4+CD127dim/−CD25+ population for each treatment shown from a representative data of three donors. **c** TGF-beta treatment increases the percentage of FOXP3+CD25+ cells in CD4+ population, shown from representative data of three donors. FMO-CD127 and FMO-FOXP3 were used as references for gating. **d** Lymphocytes from six different donors were cultured, treated, and stained as described in **a**. Percentages of FOXP3+CD25+ cells for each donor were normalized to its untreated control and plotted cumulatively. **e** Number of CD25+ on CD4+ cells is unchanged by treatments. Samples from **d** were analyzed for percentage of overall CD25+ cells and for CD25 intensity (MFI, median fluorescence intensity of CD25-labeling fluorophore). Six independent donors per group were tested. **f** TGF-beta and RAD001+TGF-beta treatments increase numbers and intensity of expression of FOXP3+ on CD4+ T cells (MFI of FOXP3-labeling fluorophore). **g** iTreg cells were generated as in **a**, enriched for CD4+CD127dim/−CD25+cells by sorting, responder cells (autologous PBMCs) labeled with CFSE and co-cultured with labeled responders at different ratios with overnight activation. Suppression ability of treated and untreated cells for all ratios shown, expressed as 1-[division index of tested cells/average division index of responders alone]. Division indices obtained from the FlowJo Proliferation Platform of tests on three independent donors per test condition. **a–g** *P* values determined by statistical analysis using two-way ANOVA tests with Dunnett post-ANOVA test determination with SEM shown. Source data are provided as a Source Data file.

activated TGF-beta, RAD001, or RAD001+TGF-beta, then at 12 d treated cells were enriched using well-established Treg cell markers (CD4+CD127dim/−CD25+) by magnetic bead sorting (see Supplementary Fig. 1 for strategy and step-by-step isolation data from one representative donor). Representative parental populations of CD4+CD127dim/−CD25+ cells (Fig. 2a) and phenotype during treatments are shown (Fig. 2b). Treatment with TGF-beta alone, RAD001 alone, or combined treatments produced similarly large increases in CD4+CD127dim/−CD25+ cells (Fig. 2b). A high percentage of FOXP3+CD25+ cells were induced by TGF-beta, both in TGF-beta alone and in double (TGF-beta+RAD001)-treated cells (Fig. 2c, d). There was a small and not statistically significant increase in the percentage of CD4+CD25+ cells with treatments which is not surprising, as untreated cells are activated effector T cells, which strongly upregulate CD25 (compare with CD25−). In contrast, FOXP3+ cells and FOXP3 intensity strongly increased in samples treated with TGF-beta or RAD001+ TGF-beta (Fig. 2f). TGF-beta, therefore, expands the FOXP3+CD25+ population of human CD4+ Treg cells, which was not observed with mTORC1 inhibition alone (compare to Fig. 2d). We also note that there was no evidence that cell sorting itself altered the percentages of FOXP3+CD25high cells nor the intensity of expressed markers.

Cells from the different treatments were compared with untreated Th0 cells for immune-suppression ability, a hallmark of Treg cell activity[37,38]. Sorted T cells were co-cultured with autologous PBMCs (responders), labeled with a division cycle tracker, seeded at different ratios of PBMC responders to effector T cells, activated overnight, and the division rate of the CD8+ responder population determined for activated dividing responder cells, measured by peak area (Supplementary Fig. 2). Composite immune-suppression data demonstrate that mTORC1 inhibition plus TGF-beta treatment generated the strongest immune-suppression activity across all responder:effector ratios, averaging approximately ninefold higher than single TGF-beta or RAD001 treatment (Fig. 2g).

To further expand these findings, we assessed by flow cytometry additional Treg cell markers, including canonical Treg cell markers such as CD101, CD103, differentiated Treg cell immunomodulatory cytokines such as TGF-beta and IL-10, and GITR, the latter a key marker of functional Treg cells[39–41]. TGF-beta plus mTORC1 inhibited (RAD001 treated) CD4+CD127dim/−CD25+FOXP3+ iTreg cells were found to concomitantly express the highest levels of GITR, CD101, CD103, TGF-beta and IL-10 (Fig. 3a). Importantly, within the Treg cell compartment (CD4+CD127dim/−CD25+FOXP3+ cells) the percentage of

CD101hi cells (Fig. 3b, c) and CD103hi cells (Fig. 3d, e; Supplementary Fig. 3a) were strongly increased by TGF-beta treatment alone or combined treatment (RAD001+TGFβ) but not by RAD001 alone (Supplementary Fig. 3b), although strong immune-suppression activity requires combined treatment. The non-Treg cell compartment, which constitutes <40% of CD4+ T cells in the post-activation population in these studies, expressed only very low levels of IL-10 and TGF-beta as expected (Supplementary Fig. 3c). Notably, αCD3/αCD28 plus IL-2 treatment of naive T cells has been shown to only give rise to iTreg cells (CD4+CD127−/lowCD25+), naive CD4+ T cells (CD4+CD25−CD45RA+), and memory T cells (CD4+CD25−CD45RA−) without the development of other T helper subsets[10,42–44]. Collectively with the data from Fig. 2, these findings show that combined mTORC1 inhibition and TGF-beta treatment produce strongly immune-suppressive CD4+Treg cells.

**Development of immune-suppressive Treg cells is associated with low levels of protein synthesis.** As a major effect of mTORC1 inhibition is downregulation of eIF4E-dependent mRNA translation, we next determined how translation is affected during the development and acquisition of immune-suppression function in Treg cells. The downregulation of mTORC1 activity required to generate Treg cells would be expected to also strongly reduce Treg cell protein synthesis. With mTORC1 inhibition, most mRNAs should be translationally downregulated, which would be apparent in mRNAs binding fewer ribosomes. mRNA-ribosome complexes (polysomes) were extracted from Treg cells generated by 12 d treatment with TGF-beta, RAD001, or RAD001+TGF-beta, and from untreated CD4+ T cells, then resolved by sucrose gradient density centrifugation (Fig. 4a). Ratios of areas underlying the polysome and monosome portions of the profiles were plotted, as indications of translation activity (Fig. 4b). Compared with untreated controls, polysome content was reduced 15–20% by TGF-beta, 30–35% by mTORC1 inhibition with RAD001, but ~70% by combined treatment of mTORC1 inhibition plus TGF-beta, which also produced the largest population of strongly immune-suppressing Treg cells. Real-time analysis of protein synthetic rates measured by puromycylation labeling of differentiating Treg cells, showed a 60–65% reduction in overall protein synthesis in CD4+ T cells (Fig. 4c, d). The reduction in protein synthesis by moderate levels of RAD001+TGF-beta is consistent with the established greater sensitivity of lymphocytes to inhibition of protein synthesis by mTORC1 inhibition than many other cell types[29]. Thus, activated

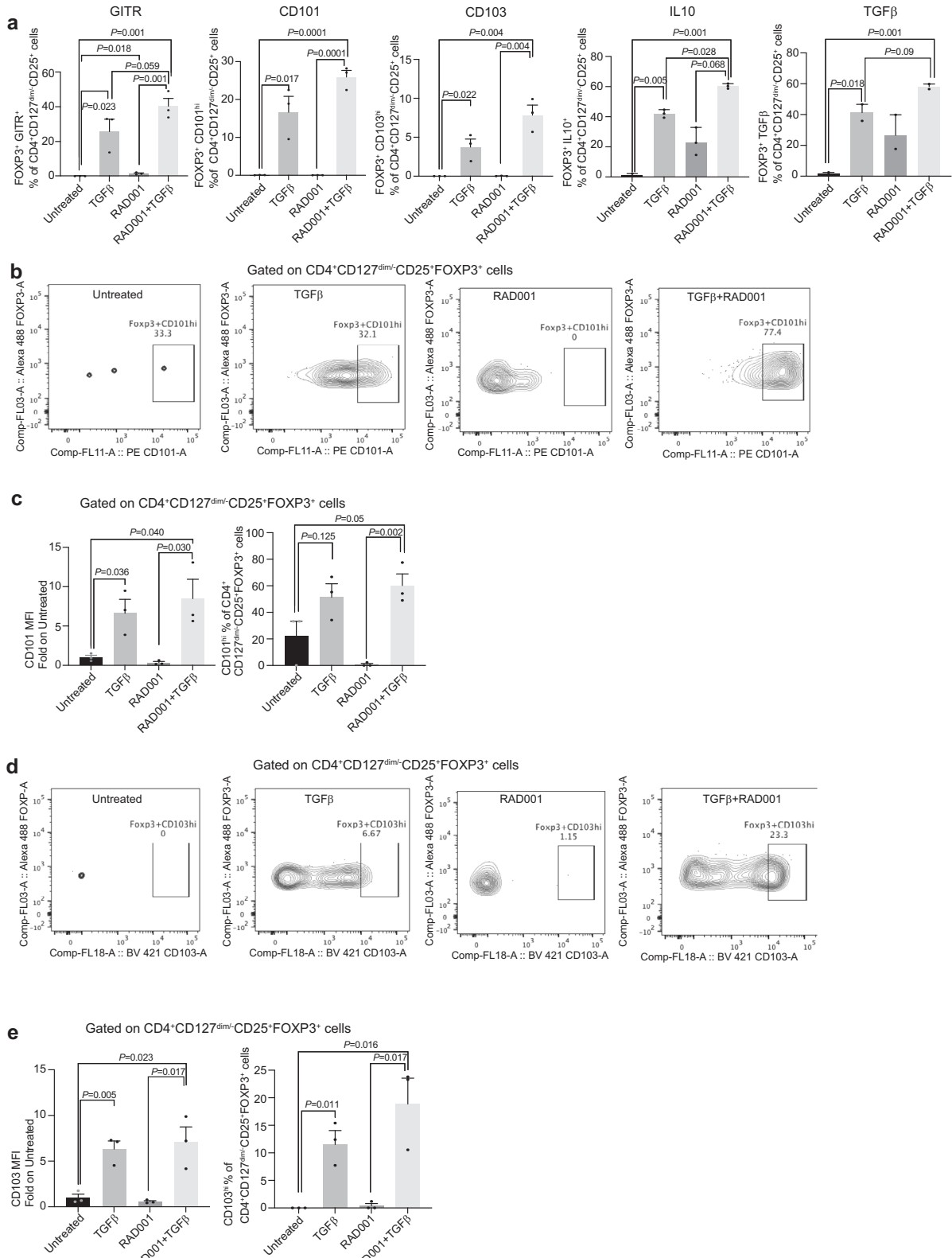

T cells differentiate into Treg cells with ~30% of the protein synthesis activity of naive CD4$^+$ T cells.

To understand how Treg cells can develop with such low levels of ongoing protein synthesis, we investigated the effects of TGF-beta and mTORC1 inhibition on the mechanism of canonical cap-dependent mTORC1/eIF4E mRNA translation initiation. mTORC1 signaling, shown by phosphorylation of mTORC1 targets P-S6 and P-4E-BP1, was most strongly blocked by RAD001 but not by TGF-beta treatment (Fig. 4e). TGF-beta alone did not impair mTORC1 activity, which therefore cannot account for the modest reduction in overall protein synthesis observed with this treatment. STAT5 phosphorylation was increased in all treated cells as expected during Treg cell induction[45], but was highest in RAD001+TGF-beta treated CD4$^+$ T cells. These data

**Fig. 3 iTreg cells differentiated by mTORC1 inhibition and TGF-beta treatment express established markers of Treg cells.** Human naive CD4$^+$ T cells were cultured and induced to differentiate to iTreg cells, then analyzed by flow cytometry as described in the legend to Fig. 2a. **a** RAD001+TGF-beta-induced CD4$^+$CD127$^{dim/-}$CD25$^+$FOXP3$^+$ iTreg cells that express high levels of GITR, CD101, CD103, TGF-beta, and IL-10. Quantitation was derived from three independent donors. *P* values were determined by statistical analysis using unpaired *t* test, with mean and SEM shown. **b** Representative CD4$^+$CD127$^{dim/-}$CD25$^+$FOXP3$^+$ cell population from three studies expressing CD101, +/− TGF-beta, +/− RAD001 treatment. Percentages normalized to untreated control for each donor. FMO-CD127 and FMO-FOXP3 were used as references for gating. **c** Fold increase in CD101 and percentage of the CD4$^+$CD127$^{dim/-}$CD25$^+$FOXP3$^+$ population obtained from three independent donor iTreg cells in **b**, normalized to each donor untreated control. **d** Representative flow cytometry of three independent donors of the CD4$^+$CD127$^{dim/-}$CD25$^+$FOXP3$^+$ cell population expressing CD103, +/− TGF-beta, +/− RAD001 treatment. Percentages normalized to untreated control for each donor. FMO-CD127 and FMO-FOXP3 were used as a reference for gating. **e** Fold increase in CD103 and percentage of the CD4$^+$CD127$^{dim/-}$CD25$^+$FOXP3$^+$ population from three donors in **d** normalized to each donor untreated control and plotted. *P* values were determined for **c**–**e** by two-way ANOVA test with Dunnett post-ANOVA test determination, with mean values and SEM shown. Source data are provided as a Source Data file.

demonstrate that the most suppressive Treg cells develop during substantial downregulation of mTORC1 activity and inhibition of protein synthesis. Therefore, to better understand the transcriptional and translational control of Treg cell development, we carried out a genome-wide transcriptomic and translatomic analysis of untreated but activated Th$_0$ effector CD4$^+$ T cells compared with RAD001, TGF-beta, or combined treated, differentiated Treg cells.

**Downregulation of mTORC1 and activation of TGF-beta reprogram the translatome for Treg cell development.** We compared the transcriptome (genome-wide mRNA levels) and translatome (genome-wide mRNA translation) of IL-2-activated CD4$^+$ T cells to differentiated Treg cells using total mRNA and well-translated mRNA (≥4 ribosomes/mRNA) fractions extracted from polysome gradients. Transcriptomic and translatomic values of mRNA levels changing in treated cells were compared to untreated cells to quantitate transcription and translation changes resulting from treatments (Fig. 5a). TGF-beta treatment alone resulted in a substantial increase in new mRNAs (Fig. 5a). Most mRNAs that were increased in abundance (increased transcription) resulted from TGF-beta treatment (Fig. 5b). In contrast, most mRNAs that were decreased in abundance resulted from RAD001 inhibition of mTORC1 activity (Fig. 5b), probably due to translational inhibition of transcription factor mRNAs. This is consistent with mTORC1 inhibition, which strongly reduced the translation of most mRNAs compared with untreated cells, shown by the increased lower spread in the translated quadrants of scatter plots (Fig. 5a). In general, mRNAs in the translatome (in polysomes) were much more strongly downregulated than their abundance (transcriptome) by mTORC1 inhibition. Combined treatment of RAD001 (mTORC1 inhibition) and TGF-beta markedly increased the number of mRNAs that were both transcribed and translated compared to single treatments (Fig. 5a, b). This suggests that TGF-beta-mediated transcription is coupled to the ability to be translated despite mTORC1 inhibition and downregulation of canonical eIF4E-dependent mRNA translation.

Selective translation of Treg cell mRNAs is supported by transcriptomic/translatomic data scatter plots (Fig. 5c; Supplementary Data 1). Log$_2$ scatter plots of independent mRNA levels and their translation indicates the major action of TGF-beta is indeed induction of the Treg cell transcriptome, whereas the major effect of mTORC1 inhibition is both inhibition of overall protein synthesis and an increase in the translation of a small number of mRNAs, which is evident in combined treatment plots. Although TGF-beta treatment alone increased translation of a small number of mRNAs, it was of lower magnitude, which was strongly elevated in RAD001+TGF-beta-treated cells. This conclusion is also supported by the analysis of RAD001-treated cells. In RAD001-alone treated cells, only very modest increases in newly translated mRNAs were observed (note lower scaling of

data). This suggested the possibility that mTORC1 inhibition of cap/eIF4E-dependent mRNA translational activity actually promotes translation of TGF-beta-induced mRNAs. Therefore, we next identified the specific genes/mRNAs that were transcriptionally induced by TGF-beta treatment and selectively translated by mTORC1 inhibition.

**Downregulation of mTORC1 and activation of TGF-beta drives selective translation of Treg cell mRNAs.** We investigated ongoing and increased translation of mRNAs under conditions of mTORC1 inhibition, TGF-beta treatment, or both in activated differentiating T cells. Full lists of upregulated and downregulated transcriptomic and translatomic mRNAs across all treatment groups were compiled (Supplementary Data 2–8). We compared genes that changed in expression and mRNAs that changed in polysome abundance, respectively, in all three treatment groups (TGF-beta, RAD001, RAD001+TGF-beta), which were subjected to top canonical pathways characterization using Ingenuity Pathways Analysis (IPA) (Supplementary Fig. 4). mTORC1 inhibition and/or TGF-beta treatment downregulated pathways involved in the determination of T helper cells other than that of Treg cells, and upregulated pathways involved in Treg cell determination, cell adhesion, and other properties associated with Treg cell function. mTORC1 inhibition plus TGF-beta treatment produced the strongest changes compared with any single treatment for both upregulated and downregulated mRNAs, in both the transcriptome and translatome (Fig. 6a, Supplementary Data 2–8). Specific mRNAs that were particularly enriched in the pool of actively translating mRNAs (≥4 ribosome polysomes) were those encoding well-established Treg cell-associated proteins (Fig. 6b). These included *CD101*, *ITGAE* (*CD103*), *FOXP3*, *CTLA-4*, *IKZF4*, *IKZF3* (*IKAROS* family *EOS* and *AIOLOS*, respectively), and *IL2RA* (*CD25*), among others. These mRNAs were all transcriptionally induced by TGF-beta and were also well translated despite RAD001-mediated mTORC1 inhibition of protein synthesis (Fig. 6b). The TGF-beta-mediated transcriptional upregulation of *CTLA-4*, *IKZF4*, *IKZF3*, and *IL2RA* was not as large as the other Treg cell mRNAs because activated but non-committed CD4$^+$ T cells also express these mRNAs. mRNA abundance and polysome enrichment changes identified for *FOXP3*, *IL2RA*, *MMP-2* (upregulated), and *IL-22* (downregulated) were validated by qRT-PCR quantitation of mRNA levels (Supplementary Fig. 5).

Interestingly, we also found a set of non-canonical Treg cell mRNAs that were strongly increased by TGF-beta treatment and efficiently translated despite mTORC1 inhibition. Most enriched were genes/mRNAs encoding *PRICKLE1*, *PTCHD1*, *IRS2*, *ATP1B1*, *SLC16A2*, *GJB6* (Fig. 6c). These non-canonical genes/mRNAs possess activities that could be involved in Treg cell differentiation and immune-suppression functions (Supplementary Table 1). For example, of the seven top non-canonical

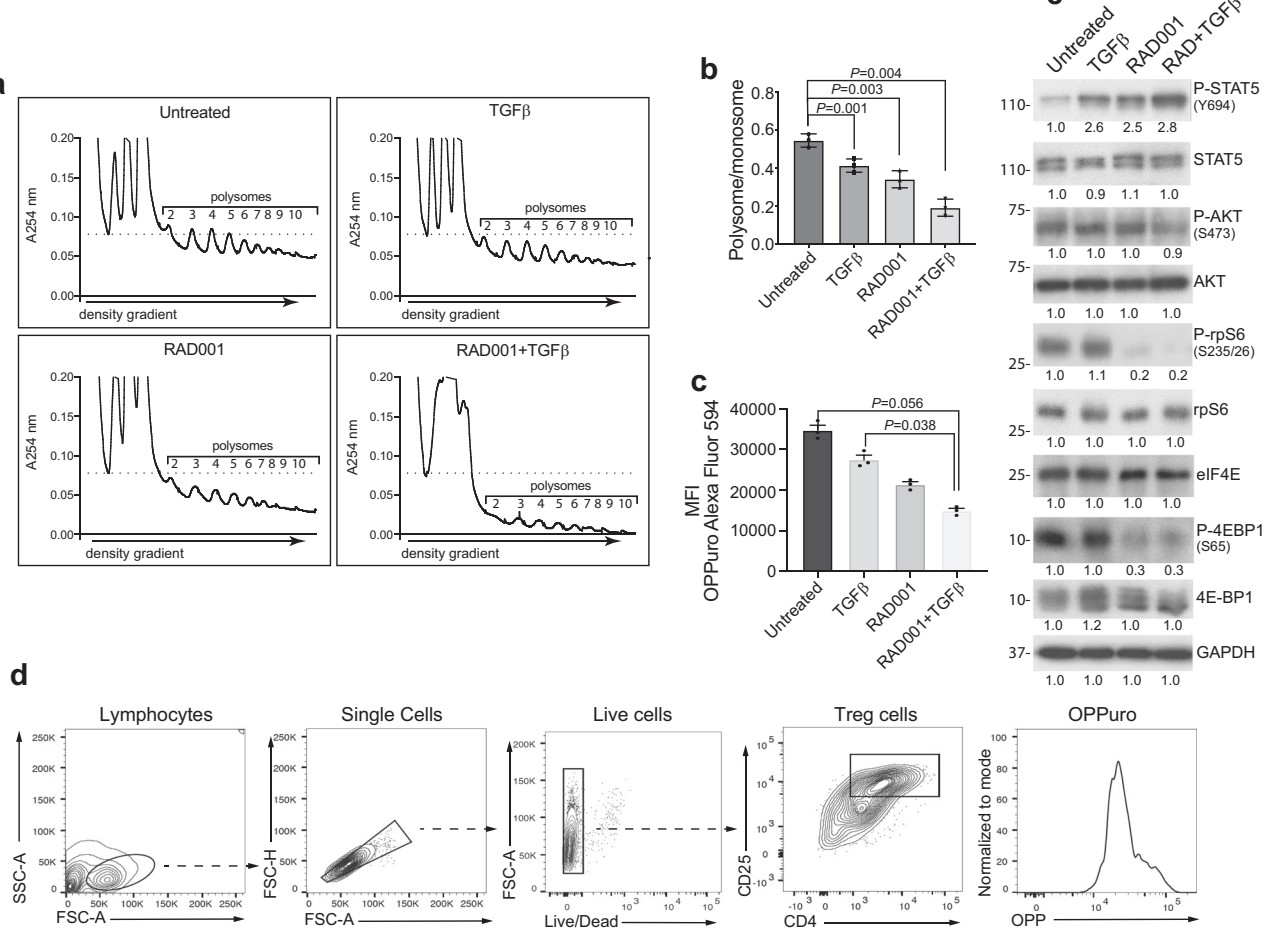

**Fig. 4 mTORC1 blockade strongly reduces protein synthesis and mTORC1 signaling activity. a** RAD001+TGF-beta-treated CD4+ Treg cells have strongly reduced polysomes in number and ribosome mRNA content, indicative of both low rates of ribosome loading and impaired protein synthesis. Translating ribosomes associated with mRNAs (polysomes) were extracted from untreated, TGF-beta-treated, RAD001-treated, and RAD001+TGF-beta-treated CD4+ T cells obtained from one donor per study, and resolved by sucrose density gradient centrifugation with continuous monitoring of RNA absorbance at 254 nm. From left to right, peaks correspond to the bulk of non-ribosomal RNA, 40 S ribosomal subunit, 60 S ribosomal subunit, 80 S ribosomes (monosomes), and polysomes (labeled). The number of ribosomes in polysomes is indicated. Polysome analysis was conducted three times from three donors. **b** Quantification of polysome levels in **a** during treatment from three donors. Areas under the curve (AUC) for monosomes and polysomes were calculated, presented as ratio of polysome to monosome integrals, with the ratio for untreated samples set at 1. *P* values determined using Fisher's exact test with means and SEM shown. **c** Untreated and RAD001+TGF-beta differentiated CD4+ Treg cells at 12 d post-differentiation from three donors were labeled by puromycylation using Click-iT Plus OPP and quantified by flow cytometry for Alexa Fluor 488 incorporation into protein for synthesis rates. Data are normalized with untreated control group mean fluorescent intensity (MFI). *P* values determined by one-way ANOVA test with Dunnett post-ANOVA test determination, with mean values and SEM shown. **d** Gating strategy for OPP-quantified protein synthesis rate analysis by flow cytometry within the Treg cell compartment. Representative of three independent studies from three donors. **e** mTORC1 pathway is downregulated in RAD001+TGF-beta-treated cells, whereas STAT5 phosphorylation is increased. Untreated and treated CD4+ T cells from four pooled donors were used for normalized representation, as described in Fig. 2 legend. Cell lysates were obtained and equal protein amounts were analyzed by immunoblot. Quantification of individual bands by digital pixel intensity is shown below each blot, normalized to untreated CD4+ T cells. STAT5 is a marker for induced Treg cells. A study with pooled specimens from two donors was independently performed twice with similar results. Source data are provided as a Source Data file.

mRNAs that demonstrated eIF4E/mTORC1-independent translation, regulation of differentiation by WNT, Sonic Hedgehog signaling and homeotic gene expression pathways (PRICKLE1, PTCHD1, SCML1, respectively) were the top representative functions, as well as gap junction protein regulators (GJB6).

**Treg cell mRNA 5′-UTRs confer translation despite mTORC1 and eIF4E inhibition**. When mTORC1 is active, 4E-BPs are hyper-phosphorylated and inactive, freeing eIF4E to recruit the translation initiation complex and the 40 S small ribosomal subunit to capped mRNAs[46]. When mTORC1 is inhibited, eIF4E

is bound to the 4E-BPs and displaced from eIF4GI or eIF4GII, blocking eIF4E-dependent mRNA translation. Translation can still take place on mRNAs that have little requirement for eIF4E and mTORC1 activity, either because they contain IRES (elements) or they utilize an alternate mechanism of cap-dependent but eIF4E/mTORC1-independent translation mediated by the DAP5/eIF3d complex[22,47].

Since translation initiation is often controlled by 5′-UTRs, we examined the 5′-UTR sequences of selectively translated mRNAs in Treg cells. mRNAs in RAD001 + TGF-beta treated Treg cells that maintain translation or increase in translation despite mTORC1 inhibition were compared to those that are

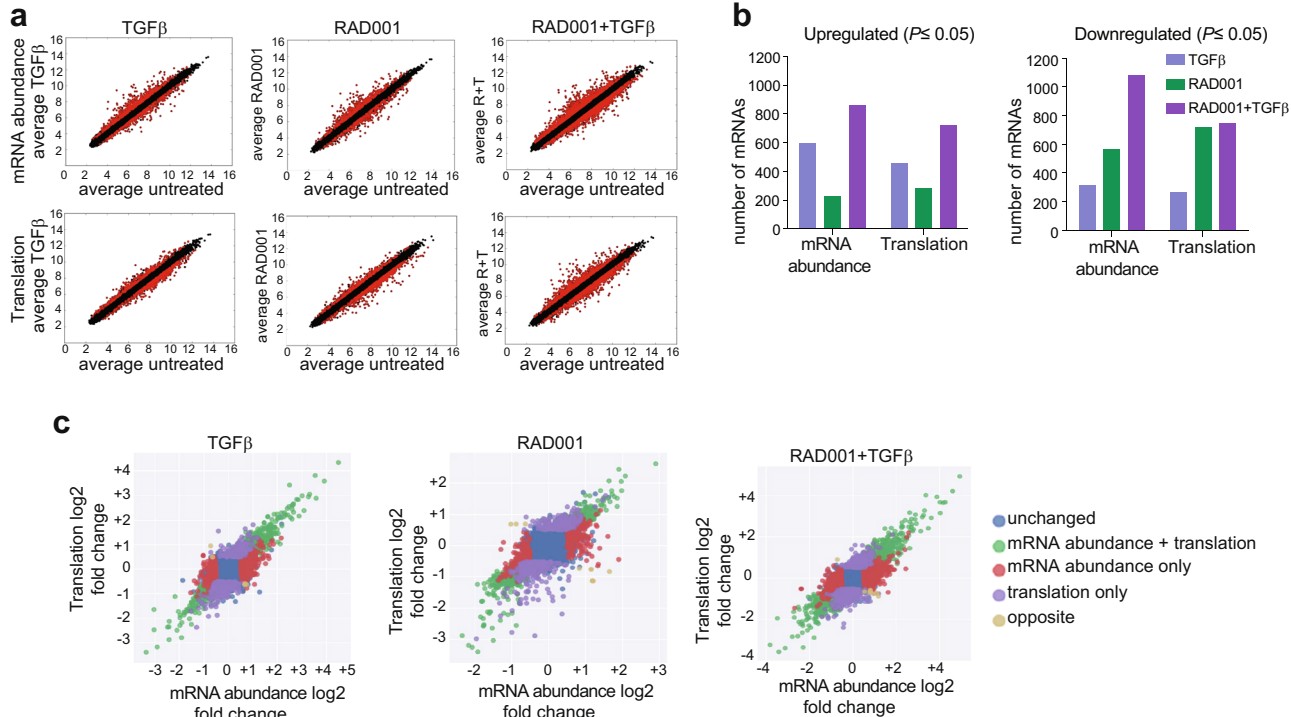

**Fig. 5 mTORC1 inhibition with TGF-beta-treatment reprograms the CD4$^+$ T-cell translatome. a** Isolation of CD4$^+$ T cells from human peripheral blood mononuclear cells (PBMCs) and treatment carried out from three independent donors as described in Fig. 2a legend. RAD001+TGF-beta treatment more strongly alters mRNA abundance (transcription) and translation than single treatment with RAD001 or TGF-beta. mRNA abundance and translation values were filtered for statistical significance ($P < 0.05$) shown in scatter plots, comparing log$_2$-transformed values for each treatment vs. untreated. Transcriptome (mRNA abundance) and translatome (translated) data sets are shown. Each dot represents one gene/mRNA, treated ($Y$ axis) versus untreated ($X$ axis). Black dots, genes/mRNAs unchanged between untreated and treated cells. Red dots, genes/mRNAs downregulated or upregulated by treatment. Results shown for three donors, two replicates each. Data analysis involved the use of GCCN and SST transformation algorithms, RMA background correction, limma R package statistics, linear regression models, Fisher's exact tests, and Benjamin–Hochberg corrections for multiple analyses. **b** Number of statistically significant ($P < 0.05$) transcriptionally or translationally upregulated or downregulated genes/mRNAs for each treatment analyzed as in **a**. RAD001+TGF-beta treatment relieves transcriptional inhibition of a pool of downregulated genes through translation, and translationally represses a set of genes upregulated in transcription. **c** Log$_2$ scatter plots of transcriptomic and translatomic results for all treatment groups, comparing log$_2$ changes of translation to mRNA abundance (transcription) for each group. Translation only (translation efficiency) compares translation levels normalized to mRNA levels: mRNA abundance + translation is not normalized to mRNA levels; opposite represents a small number of mRNAs whose translation level is the opposite of the change in mRNA abundance. Source data are provided as Supplementary Data 1–8.

translationally downregulated or inhibited. We searched for motifs and elements using GenomeScope and MEME bioinformatics engines. For the canonical mRNAs, we were not able to identify any sequences, motifs, or computationally predicted secondary structures in the 5′-UTRs that are in common, and are not also found in translationally downregulated mRNAs, apart from a non-statistically significant tendency for potentially relaxed cap-proximal secondary structure, which has been shown to reduce the requirement for mTORC1 activity and eIF4E[48]. For the non-canonical Treg cell mRNA 5′-UTRs, the SCOPE algorithm detected a series of low complexity GC-rich motifs (Supplementary Fig. 6a), which are significantly enriched in some of the translated compared with downregulated mRNAs ($Z$ score statistical analysis, 67.1; $P < 0.00001$ by two-tailed $t$ test) (Supplementary Fig. 6b, c, shown for the *PRICKLE* mRNA 5′-UTR). Notably, it remains unknown how the eIF4E/eIF4GI and DAP5/eIF3d complexes identify and discriminate 5′-UTRs of respectively translated mRNAs.

We asked whether an alternate mechanism of cap-dependent mRNA translation that does not use eIF4E nor requires mTORC1 activity is responsible for continued translation of Treg-cell development mRNAs. We also asked whether the 5′-UTRs of selectively translated canonical and non-canonical Treg cell mRNAs confer translation capability despite mTORC1 inhibition. Human

Treg cells were generated from naive human CD4$^+$ T cells from four donors using TGF-beta and RAD001 as described for Fig. 1. Following treatments, donor cells were pooled for analysis to normalize against individual variation and subjected to immunoblot for pre-initiation complex and translation regulatory proteins (Fig. 7a). There was no change in expression between activated and Treg cell differentiated cells in levels of eIF4E, eIF4A, 4E-BP1 or 2, and β-actin control, there was a strong decrease in eIF4GI levels, and a strong increase in DAP5 and eIF3d levels in differentiated Treg cells. Next, human naive CD4$^+$ T cells were treated with TGF-beta, RAD001, or both from four independent donors, and equal amounts of protein extracts were subjected to m$^7$GTP cap-chromatography (Supplementary Fig. 7a). In CD4$^+$ T cells treated with RAD001, with or without TGF-beta, cap-bound eIF4E showed greatly diminished binding to eIF4GI and increased binding to 4E-BP1 and −2. These data show that mTORC1 inhibition in Treg cells strongly downregulates the canonical eIF4E-eIF4GI complex that promotes cap-dependent mRNA translation, and suggest that TGF-beta upregulation of DAP5 and eIF3d might promote an alternative mechanism for selective translation of Treg cell differentiation and function mRNAs.

We therefore compared eIF3d and eIF4E binding to m$^7$G cap structures using cap (m$^7$GTP) chromatography and silencing of eIF4E or eIF3d. Human HEK 293 T cells were used because they

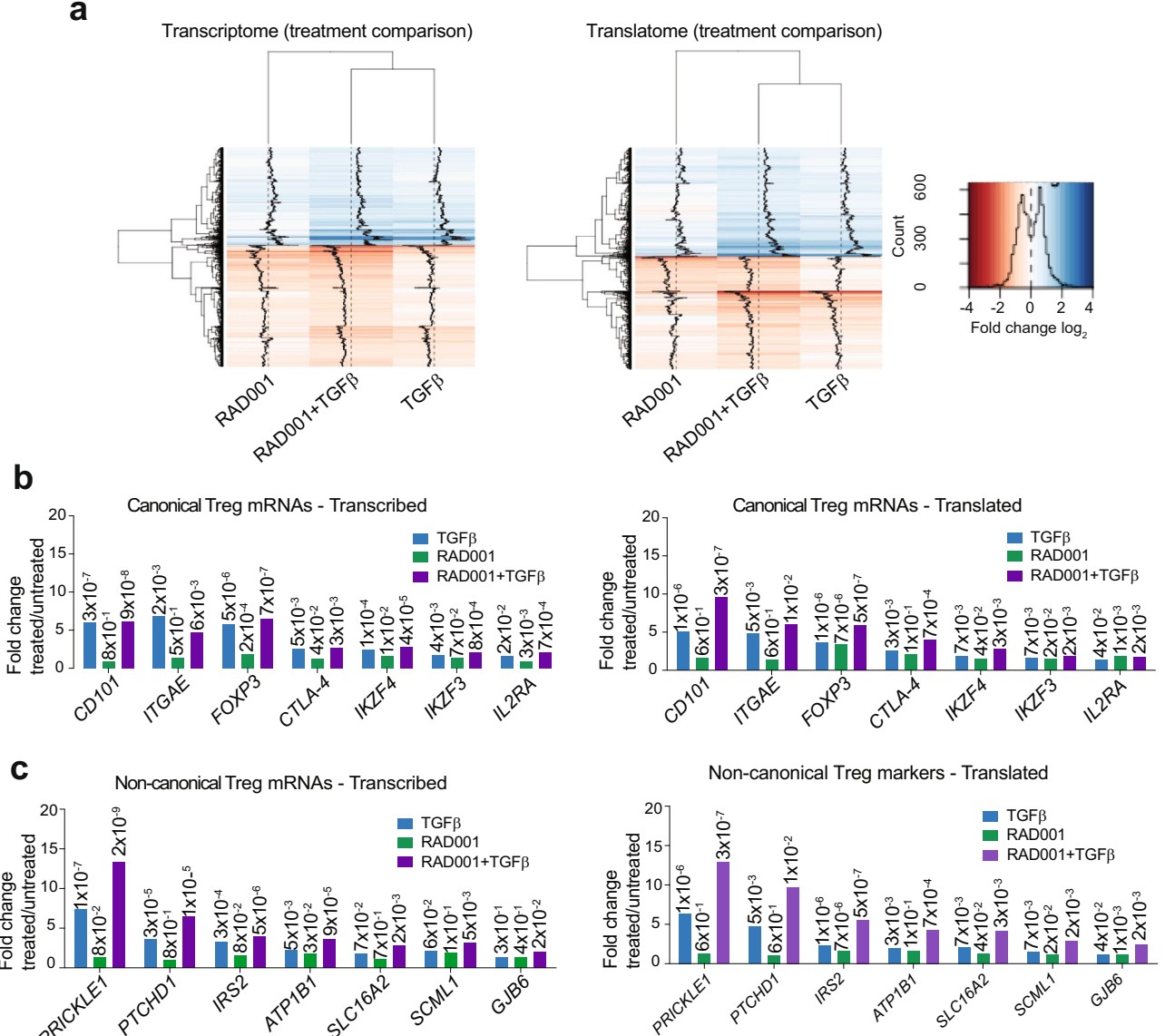

**Fig. 6 mTORC1 inhibition and TGF-beta treatment selectively promotes translation of canonical Treg cell mRNAs. a** Heat map of genes and mRNAs upregulated or downregulated in CD4⁺ T cells treated with RAD001+TGF-beta compared with single treatments. RAD001+TGF-beta treatment increases the extent of upregulation or downregulation of each gene/mRNA compared with single treatments. Log₂ of fold-changes of genes/mRNAs in TGF-beta, RAD001, and RAD001+TGF-beta compared with untreated controls were used to generate heat maps for genome-wide mRNA abundance (left panel) and translation of mRNAs (right panel). Each row corresponds to a gene/mRNA. Blue indicates upregulation, red downregulation. Darker shades correspond to higher log₂ fold-change values. **b** Treg cell canonical mRNA levels are upregulated by TGF-beta and increased in translation by mTORC1 inhibition. Transcription and translation levels of genes/mRNAs characterizing the Treg cell population are shown for each treatment, each bar representing the fold-change against untreated. *P* values are shown on top of each histogram derived from genome-wide analysis (analyzed filtered source data provided in Supplementary Data 2–8). **c** Non-canonical mRNAs with activities associated with Treg cell development or function are increased in expression by TGF-beta and translation by mTORC1 inhibition. IPA software analysis tools calculated *P* values as shown above each histogram. **a** Data represent compiled analysis of genome-wide data from three donors, two replicates each. Data analysis (**b** and **c**) by IPA used a suite of bioinformatics and statistical analysis tools, including right-tailed Fisher's exact tests to determine *P* values and Benjamin–Hochberg corrections for multiple analyses.

do not require TGF-beta treatment to express higher levels of DAP5 and eIF3d, they are easily and efficiently transfected with small interfering RNA (siRNAs), and they can produce in vitro translation extracts to test Treg cell mRNA 5′-UTRs for DAP5, eIF3d and eIF4E translation requirement, which is not possible with human CD4⁺ T cells. eIF4E, eIF3d or DAP5 mRNAs were silenced using previously validated siRNAs[22,49]. Silencing at these levels did not induce toxicity or evidence of cell death (Supplementary Fig. 7b, c). Both eIF4E and eIF3d were detectably retained by m⁷GTP caps and silencing of eIF3d abolished its

detectable cap binding. Interestingly, silencing eIF4E by ~50% increased eIF3d cap binding (Fig. 7b), indicating that eIF4E under these conditions is more strongly competitive than eIF3d in m⁷G cap binding activity.

The 5′-UTRs of representative CD101 and CD103 Treg mRNAs were inserted in a Renilla luciferase reporter expression vector. The 5′-UTR of the integrin β1 mRNA (*ITGβ1*) was used as a control because it was found previously to be highly eIF4E dependent[50], and was translationally downregulated by RAD001 in our dataset. Luciferase mRNAs were generated in vitro by T7-directed

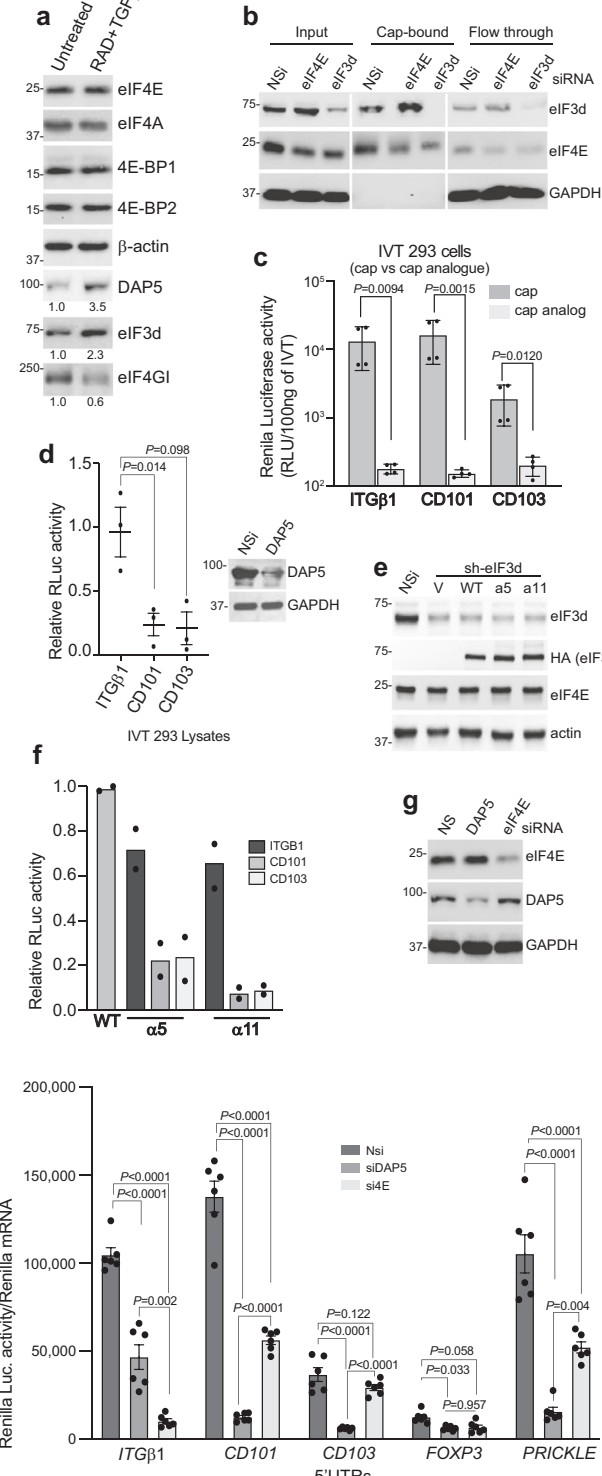

**Fig. 7 Treg cell mRNAs are translated in a DAP5/eIF3d cap-dependent manner. a** Human naive CD4$^+$ T cells from four different donors were differentiated from PBMCs into Treg cells with TGF-beta and RAD001, then isolated as described in Fig. 2a legend. Equal protein amounts of cell lysates from each donor Treg cell population were pooled and subjected to immunoblot analysis as shown, comparing IL-2 activated but otherwise untreated CD4$^+$ T cells prior to differentiation to their post-differentiated Treg cells. Quantification of individual bands by digital pixel intensity is shown below each blot, normalized to untreated CD4$^+$ T cells. Results shown are representative of two independent studies from different donors. **b** HEK 293 cells were transfected with non-silencing (Nsi) siRNAs or siRNAs to eIF4E or eIF3d for 72 h, cytoplasmic protein extracts prepared and equal amounts subjected to m$^7$G-Sepharose cap-chromatography. Input, cap-bound, and unretained flow-through proteins were identified by immunoblot analysis. Representative immunoblots are shown of two independent studies from two donors each. **c** In vitro translation extracts were prepared from 293 T cells[22] (see Methods). Renilla luciferase mRNAs containing the 5′-UTR of ITGβ1, CD101 or CD103 mRNAs were generated by in vitro transcription with T7 RNA polymerase containing an m$^7$GTP cap structure or non-functional cap analog (ApppG) and polyadenylated. Equal amounts of mRNAs were used to program translation extracts, and relative luciferase activity was determined in four independent experiments. **d** Translation extracts prepared as above from Nsi or DAP5 silenced 293 T cells and programmed with equal amounts of m$^7$GTP-capped Renilla luciferase mRNAs containing the 5′-UTR of ITGβ1, CD101, or CD103 mRNAs. Luciferase activity was determined from three independent experiments, repeated twice showing mean values. Immunoblot of Nsi and DAP5 silenced 293 T cell extracts, representative of two experiments. **e** 293 T cells were engineered to stably express doxycycline (Dox) inducible non-silencing (Nsi) or eIF3d silencing shRNAs, Dox-induced for 72 h, then cells transfected with expression vectors for HA-tagged WT eIF3d (HA), or cap-binding deficient mutants α5 or α11. Equal protein amounts of cell extracts were subjected to immunoblot. Representative immunoblots shown from two independent studies. **f** 293 cell translation extracts were prepared and programmed as above with m$^7$GTP-capped Renilla luciferase reporter mRNAs containing the 5′-UTR of ITGβ1, CD101 or CD103. Luciferase activity determined in triplicate from two independent studies. IRES-driven Renilla luciferase was used for normalization. **g** 293 cells were transfected with Nsi, DAP5 or eIF4E siRNAs for 72 h. Equal protein amounts of cell extracts were subjected to immunoblot. Representative immunoblots shown from two independent studies. **h** 293 cells silenced as above were transfected with plasmids expressing Renilla luciferase reporter mRNAs containing the 5′-UTR of ITGβ1, CD101, CD103, FOXP3, or PRICKLE mRNAs. Light intensity values shown for Renilla luciferase activity normalized to Renilla mRNA determined by qRT-PCR. Mean values of triplicate assays of six independent experiments shown with SEM. (**c**, **d**, **h**) P values determined by statistical analysis using the one-way ANOVA test with Tukey's multiple comparison test determination. Source data are provided as a Source Data file.

were silenced for DAP5 and programmed with m$^7$GTP capped and polyadenylated Renilla luciferase expression reporter mRNAs. Silencing DAP5 significantly impaired translation of CD101 and CD103 5′-UTR containing mRNAs, but not those containing the ITGβ1 5′-UTR (Fig. 7d).

To determine whether the contribution of eIF3d in Treg cell mRNA translation is through its cap binding function or as a component of eIF3, in vitro translation extracts were prepared from either wild type 293 T cells or cells silenced for endogenous eIF3d, but were engineered to overexpress either of two different eIF3d mutants that lack cap-binding activity but retain wild type levels of RNA binding as part of the eIF3 complex[49]. In vitro translation directed by the ITGβ1 5′-UTR was unaltered by replacement of wild type eIF3d with either cap-binding mutant,

transcription containing either a functional m$^7$GTP cap or a non-functional ApppG cap analog and were 3′ polyadenylated. Equal amounts of each mRNA were used to program 293 T cell in vitro translation extracts. Inclusion of a non-functional cap analog abolished translation activity of all mRNAs, demonstrating that Treg cell CD101 and CD103 mRNAs are cap-dependent yet able to translate despite 4E-BP inhibition of eIF4E in differentiating Treg cells. To test whether CD101 and CD103 mRNAs instead use the DAP5/eIF3d complex, in vitro translation extracts from 293 T cells

whereas translation directed by the *CD101* or *CD103* 5′-UTRs was strongly reduced (Fig. 7e, f). Therefore, unlike the control eIF4E/mTORC1-dependent *ITGβ1* 5′-UTR, translation of representative Treg cell mRNAs, *CD101* and *CD103* both require eIF3d cap-binding activity and DAP5.

We also tested 5′-UTR translation activity for eIF4E or DAP5 dependence in transfected control and silenced cells. Translation of mRNAs controlled by the *ITGβ1* 5′-UTR were reduced 10-fold with *EIF4E* silencing, and but only twofold with DAP5/*eIF4G2* silencing (Fig. 7h). Thus, the *ITGβ1* mRNA 5′-UTR shows a strong requirement for the canonical eIF4E pre-initiation complex and weak use of the DAP5/eIF3d complex. In contrast, translation of mRNAs controlled by the *CD101* 5′-UTR were reduced by approximately ninefold by DAP5/*eIF4G2* silencing, but only about twofold by *EIF4E* silencing, indicating much greater dependence on DAP5 than eIF4E for translation. Similarly, the *CD103* 5′-UTR also showed strong dependence on DAP5 and weaker dependence on eIF4E. This was also observed for the non-canonical *PRICKLE* 5′-UTR. We also examined the translational requirements for *FOXP3* mRNA. FOXP3 is expressed in activated CD4$^+$ T cells and in Treg cells, conditions that are both eIF4E and DAP5 requiring, and was transcriptionally but not translationally increased in Treg cells. Interestingly, mRNAs translationally controlled by the *FOXP3* 5′-UTR were reduced by half with either *EIF4E* or DAP5 silencing, indicating dual usage of either mechanism, consistent with expression in both uncommitted activated CD4$^+$ T cells and differentiated Treg cells. This has been previously observed for mRNAs that can use either mechanism[22]. These data demonstrate that translation resistance to mTORC1/eIF4E inhibition is conferred in part through the 5′-UTRs of representative canonical and non-canonical mRNAs, which can also use to differing extents, the DAP5/eIF3d mechanism for cap-dependent mRNA translation.

**DAP5 silencing impairs iTreg-cell differentiation from human naive CD4$^+$ T lymphocytes.** Having demonstrated in HEK 293 cells that the 5′-UTR of Treg cell lineage differentiation and function mRNAs require DAP5, eIF3d, and an m$^7$G cap, we assessed whether DAP5 silencing impairs human iTreg-cell differentiation from naive CD4$^+$ T cells. Uncommitted T cells are very difficult to transfect or to be transformed by viral vectors. However, we found that freshly isolated human naive CD4$^+$ T cells could be effectively silenced for DAP5 expression by repeated treatment with a pool of Accell SMARTpool siRNAs targeting DAP5 mRNA or a non-targeting siRNA pool. CD4$^+$ T cells in serum-free X-Vivo 15 media were transfected three times 3 d apart while stimulated with IL-2 and treated with TGF-beta, RAD001 or both (Fig. 8a). CD4$^+$CD127$^{dim/−}$CD25$^+$FOXP3$^+$ GITR$^+$ T cell levels were determined on d 12 by flow cytometry. Cell uptake of siRNAs was verified in parallel by flow cytometry for expression of a GFP non-targeting siRNA control (Fig. 8b; Supplementary Fig. 8a). DAP5 levels were decreased by ~70%, assessed by immunoblot of equal protein amounts of non-differentiated and differentiated cells (Fig. 8c). The fraction of CD4$^+$CD127$^{dim/−}$CD25$^+$FOXP3$^+$ GITR$^+$ iTreg cells in the population of the non-silenced control and DAP5 silenced CD4$^+$ T-cell compartment (Fig. 8d; Supplementary Fig. 8b) was determined for multiple independent donor isolates (Fig. 8f). Silencing DAP5 reduced by half the induction of FOXP3$^+$GITR$^+$ Treg cells by TGF-beta and RAD001 within the CD4$^+$CD25$^+$CD127$^{dim/−}$ cell population, to baseline levels of iTreg cells found in untreated CD4$^+$ T cells (Fig. 8f, left panel). DAP5 silencing also blocked induction of Treg cells by TGF-beta and RAD001 within the CD4$^+$ T cell pool, and of live cells by half

(Fig. 8f, middle and right panels). Viability of untreated and treated cells was not affected by DAP5 silencing (Fig. 8e). Whether DAP5 silencing reverses differentiation of Treg cells could not be tested, because mature differentiated Treg cells could not be silenced by any method we tested.

## Discussion

mTOR is a sensor of physiological stress and stimulatory signals in the tissue microenvironment, and a central gatekeeper of cellular metabolism. mTOR has also emerged as a key regulator of T-cell function, plasticity and fate-determination[6,51], but its regulation of T-cell development and function through its effects on translational control are only beginning to be understood. mTOR activity has been shown to promote helper T-cell differentiation and determine Th1 or Th2 cell type through the selective activity of mTORC1 or mTORC2[17]. mTOR also promotes CD8$^+$ effector T-cell development, in part by controlling the expression of two fate-determining transcription factors[52,53], through translationally controlled mechanisms[54]. In our work, we addressed the conundrum that while reduced or inhibited mTORC1 activity skews CD4$^+$ T-cell development to FOXP3$^+$ Treg cells[16,31,55,56], both mTORC1 activity and eIF4E availability, which is controlled by mTORC1, are required for the translation of most mRNAs.

We found that inhibition of both mTORC1 and mTORC2 blocks Treg cell development. This finding likely clarifies why deletion of Raptor, an essential component required for mTORC1 activity, results in inhibition of Treg cell proliferation[20]. Raptor deletion may dysregulate rather than inhibit mTORC1 signaling and hyperactivate mTORC2[57]. In this regard, it is interesting to note that Treg cells, in particular, are less sensitive to oxidative stress than other T-cell types and are able to develop in metabolically poor microenvironments[58,59]. mTORC1 is inhibited when oxygen, energy and nutrient levels decrease[60], conditions that occur locally when the immune system responds to infection or inflammation events[61]. Inhibition of overall protein synthesis during Treg cell development in oxygen and nutrient-stressed environment which impairs mTORC1 signaling, would therefore require a mechanism that allows privileged mRNA translation despite cell stress, thus enacting peripheral Treg-cell development and immune-suppression activity. DAP5/eIF3d-mediated cap-dependent mRNA translation likely exists for this reason[22,47], and is employed by CD4$^+$ T cells to enable Treg-cell development under challenging conditions.

Our studies therefore provide an understanding of the complex relationship between T-cell metabolism, micro-environmental stress and cytokines in the determination and plasticity of T-cell fate. The ability of T cells to undergo transcriptional reprogramming by TGF-beta, coordinated with translational reprogramming by mTORC1 inhibition and DAP5/eIF3d-mediated cap-dependent mRNA translation, cued by environmental and metabolic stress, provides a mechanism for maintaining plasticity and T-cell fate-determining response with acute sensitivity, and allows rapid switching among T-cell subsets. Although many of the features of flexible T-cell reprogramming remain to be determined, particularly how metabolic and nutritional status alter phenotype, our findings demonstrate that mRNA translational reprogramming controlled by mTORC1 activity and local cytokines provides a unifying mechanism that incorporates metabolic, nutritional and cytokine responses that determine T-cell fate.

Although the activity of mTORC1 affects development of different T-cell types, Treg cell differentiation is not merely the default response to mTORC1 inhibition. Rather, mTORC1 inhibition is the basis for selective translation reprogramming of TGF-beta-induced mRNAs that possess a significantly reduced

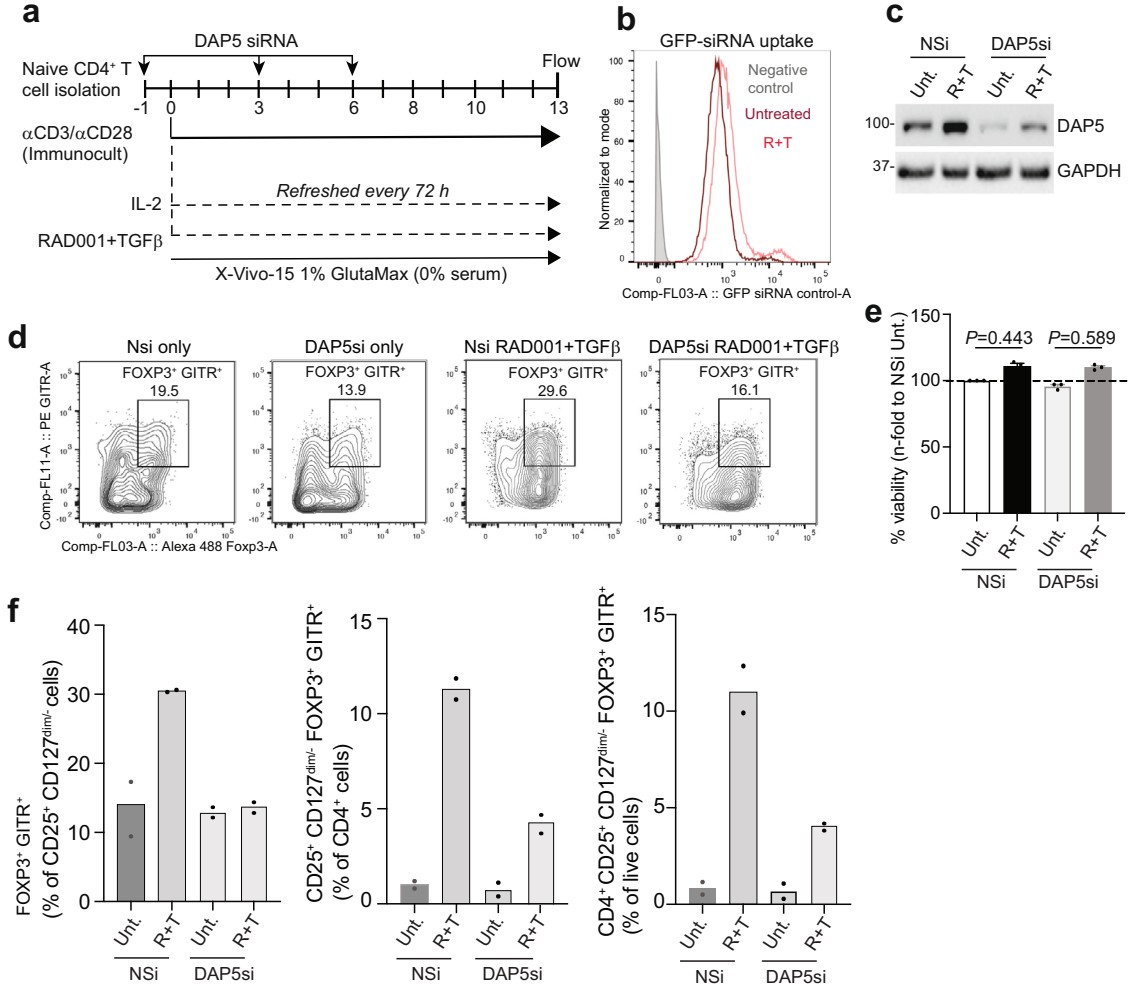

**Fig. 8 DAP5 silencing blocks human naive CD4+ T cell differentiation into Treg cells. a** Schema for silencing DAP5 in activated CD4+ T cells during differentiation. Human naive CD4+ T cells from two different donors were isolated from PBMCs as described in Fig. 2a legend. Cells were treated with 1 µM of Accell SMARTpool siRNAs targeting DAP5 or a non-targeting siRNA pool in serum-free X-Vivo 15 media containing 1% GlutaMAX for 24 h, activated with IL-2 and differentiated with TGF-beta and RAD001 to induce iTreg cells and maintained as described in Methods. Accell SMARTpool siRNAs were re-added to culture medium on d 3 and 6 after activation and cells submitted to flow cytometry on d 13. Studies included a GFP-siRNA non-targeting control to measure uptake efficiency by flow cytometry. **b** Levels of GFP uptake measured by flow cytometry in negative control (no GFP), IL-2 activated but otherwise untreated, and RAD001+TGF-beta treated (R+T) CD4+ T cells. Representative plot of two independent studies. **c** Equal protein amounts of cell lysates from two donors obtained as in **a** were pooled and subjected to immunoblot analysis. Samples correspond to untreated and RAD001 + TGF-beta treated cells. DAP5 protein levels were reduced by 70–80% with silencing. Immunoblots are representative of two independent experiments. **d** Human CD4+ T cells treated as in **a** were subjected to flow cytometry for differentiated iTreg cells (CD4+CD127dim/−CD25+FOXP3+ GITR+) determined on d 13. Representative flow plots of two independent experiments shown indicating a 51% reduction in differentiated iTreg cells to baseline levels of Treg cells isolated from PBMCs prior to differentiation. **e** iTreg cells were tested for viability by Trypan Blue exclusion assay. *P* values determined using Fisher's exact test with means and SEM shown. There is no statistically significant difference in viability between pairs of non-silenced and DAP5 silenced matched conditions. **f** Percentages of CD4+CD127dim/−CD25+FOXP3+ GITR+ cells for each of two donors were normalized to its untreated control and plotted cumulatively for IL-2 activated but otherwise untreated, and RAD001+TGF-beta treated (R+T) CD4+ T cells. Source data are provided as a Source Data file.

requirement for mTORC1 and eIF4E. It is important to note that mTORC1 downregulation itself is insufficient to mediate the most functionally effective immune-suppressive Treg cells. We found that immune-suppression activity also requires transcriptional reprogramming by exposure to TGF-beta. Thus, transcriptional and translational reprogramming act in concert to determine the Treg cell phenotype, which raises the mechanistic possibility that there are other transcriptionally and translationally aligned programs that determine other T-cell phenotypes in response to specific metabolic states and cytokines. mTORC1 activity is therefore not just a rheostat that determines the level of

eIF4E-dependent mRNA translation, but rather, acts as a switch for mRNA translational reprogramming.

There are different mechanisms of alternate translation initiation that have been described that are likely employed in different cell physiological contexts, including eIF4E, DAP5/eIF3d and eIF3d mechanisms of cap-dependent mRNA translation, as well as cap-independent IRES-mediated internal ribosome entry initiation. All these mechanisms are poorly characterized in immune cell determination. In the case of peripheral Treg cell maturation, it makes good sense that Treg cell mRNAs still utilize eIF4E/eIF4GI and DAP5/eIF3d, but have

greater dependence on DAP5/eIF3d. DAP5-mediated mRNA translation may also be compensated by the increased expression observed for both proteins during Treg cell differentiation. Moreover, there may be other factors involved in development of T-cell types as well that remain to be characterized. For instance, increased expression of certain subunits of the multi-protein translation initiation factor eIF3 have been shown to bind mRNA directly and promote translation without a requirement for eIF4E[62], and eIF3d itself was recently shown to promote translation of several hundred mRNAs[63]. In summary, our study highlights that there are alternative translation mechanisms that are important in determining T cell lineage and function that remain to be characterized.

## Methods

**Experimental models, human, and animal subject details.** Human naive T cells. PBMCs were obtained by density gradient centrifugation (Ficoll Plaque PLUS, GE Healthcare) of buffy coats purchased from the New York City Blood Bank obtained commercially from de-identified donors who were consented by the Blood Bank to donate blood. Donors were de-identified by the Blood Bank which provides PBMCs as a commercial entity, and as such no Institutional Review Board approval is required.

**Cell culture and cell treatments.** Human naive CD4[+] primary lymphocytes were isolated from PBMC by magnetic sorting using the naive CD4[+] T Cell Isolation Kit II (Miltenyi). In all, $2 \times 10^5$ cells were incubated with Blue LIVE/DEAD Fixable Dead Cell Stain kit (Life Technologies, Cat. #L-23105) in 1× PBS, then diluted in Stain Buffer (BD Pharmingen, Cat. #555346) and stained with FITC Mouse anti-human CD4 (BD Pharmingen, Cat. #555346) and PE Mouse anti-human CD45RA (BD Pharmingen, Cat. #555489). Between 1 and $1.5 \times 10^6$ isolated cells/ml were cultured in X-Vivo 15 culture medium (Lonza, Cat. #04-418Q), 1% GlutaMAX supplement (Gibco, Cat. #35-050-061) and heat-inactivated 5% human AB serum (Sigma Millipore, Cat. #4522, Lot #SLBT0317). For experiments involving treatment-free Accell SMARTpool siRNAs, serum-free X-Vivo 15 media containing 1% GlutaMAX was used. One d after isolation, naive cells were activated by plate-bound anti-CD3 and soluble 1 µg/ml anti-CD28 (Affymetrix eBioscience, Cat. #16-0037, #16-0288, respectively) in the presence of 150 U/ml IL-2 (Miltenyi, #130-093-907). Anti-CD3 coated plates were prepared by incubating a solution of 10 µg/ml anti-CD3 in 100 mM Tris HCl, pH 7.5 for 16 h at 4 °C. Treatments were added during activation and maintained at all times, and unless otherwise stated were: 2 µg/ml TGF-beta (Miltenyi, Cat. #130-095-067), 1 µM PP242 (Chemdea, Cat. #CD0258), 20 nM RAD001 (LC Laboratories) or combinations as indicated. Untreated cells were added with equal volumes of sterile DMSO. Three d after activation, cells were diluted at $1$–$1.5 \times 10^6$ cells/ml and transferred in new uncoated plates. Cells were then maintained below $5 \times 10^6$ cells/ml and IL-2 was refreshed every 3 d. Seven d after activation, a second round of activation with plate-bound αCD3 and soluble αCD28 was performed overnight. Cells were collected 12 d after the first round of activation.

**iTreg-cell enrichment.** Treated primary human CD4[+] T lymphocytes were enriched on Treg cells on d 12 by magnetic sorting using the CD4[+]CD25[+]CD127[dim/−] Regulatory T Cell Isolation Kit II (Miltenyi, Cat. #130-094-775) with some modifications of the protocol as indicated here. Magnetic beads for depletion of non-CD4[+] and CD127[high] cells were reduced at ¼ of the suggested volume, while beads for CD25[+] enrichment were doubled. Cells were resuspended at $1.5 \times 10^6$/ml in complete RPMI + 150 U/ml IL-2 and appropriate treatment and rested overnight. $2 \times 10^5$ cells were stained to check for Treg cell markers, $1 \times 10^5$ cells were used for suppression assay, and the bulk of the cells were incubated with 100 µg/ml cycloheximide 5 min at 37 °C, centrifuged at $500 \times g$ at room temperature, washed with sterile room temperature 1× PBS + 100 µg/ml cycloheximide, flash-frozen in liquid nitrogen and stored at −80 °C for polysome profile analysis.

**Flow cytometry analysis.** In all, $1$–$2 \times 10^5$ cells were incubated with Blue LIVE/DEAD Fixable Dead Cell Stain kit (Life Technologies, Cat. # L-23105) or Zombie Aqua Fixable Viability Kit (BioLegend, Cat. # 423102) in 1× PBS, and then stained with surface antibodies diluted in Stain Buffer (BD Pharmingen, Cat. #554656): FITC mouse anti-human CD4 (BD Pharmingen, Cat. #555346), APC mouse anti-human CD25 (BD Pharmingen, Cat. #555434), PE-Cy7 mouse anti-human CD127 (Invitrogen BD Pharmingen, Cat. #5 25-1278-42), PE mouse anti-human CD101 (BioLegend, Cat. #331012), Brilliant Violet 421 mouse anti-human CD103 (BioLegend, Cat. #350213) and PE mouse anti-human GITR (BioLegend, Cat. #371203). After surface marker staining, cells were fixed and permeabilized using Human FOXP3 Buffer Set (BD Pharmingen, Cat. #560098) according to manufacturer instructions and stained with BioLegend FITC mouse anti-human TGF-beta (Cat. #349605), Brilliant Violet 421 mouse anti-human IL-10 (Cat. #501421)

antibodies, and BD Pharmingen PE mouse anti-human FOXP3 (#560046) or Alexa Fluor 488 mouse anti-human FOXP3 (Cat. #560047) antibodies. Cells were added with 1% formaldehyde in Stain Buffer and read at BD LSRII flow cytometer (BD Biosciences) at the NYU Cytometry & Cell Sorting Laboratory facility. Flow cytometry data were analyzed by FlowJo software. In experiments assessing intracellular cytokine expression, cells were incubated with Activation Cocktail containing Brefeldin A (BioLegend, Cat. #423303) for 5 h at 37 °C and 5% $CO_2$ prior to flow cytometry staining.

**OPP puromycin assay.** OPP puromycin assay (Click-iT Plus OPP Alexa Fluor 594 Protein Synthesis Assay Kit, Life Technologies, Cat. #C10457) was performed following the manufacturer's instructions in primary human CD4[+] T lymphocytes treated with TGF-beta and/or RAD001 and activated as detailed above. Two different donors were used. In brief, treated cells at d 12 of differentiation were incubated with component A of the kit for 30 min at 37 °C and 5% $CO_2$, and then fixed with 3.7% formaldehyde and permeabilized with 0.5% Triton X-100. Click-iT Plus reaction cocktail was added to the samples and incubated 30 min at room temperature, protected from light. After washing, cells were stained with FITC mouse anti-human CD4 (BD Pharmingen, Cat. #555346) and APC mouse anti-human CD25 (BD Pharmingen, Cat. #555434), both diluted in Stain Buffer (BD Pharmingen, Cat. #554656). Cells were analyzed by flow cytometry as detailed above.

**Immune-suppression assay.** Suppression assay was performed as previously reported[24,64]. In brief, responder cells were autologous PBMCs thawed and labeled with CellTrace CFSE Cell Proliferation kit (Life Technologies, Cat. #C34554) according to manufacturer instructions. V-bottom 96 well plates, anti-CD3 coated, were added with $1 \times 10^5$ CFSE-labeled PBMCs suspended in 100 µl AIM-V medium (Gibco-Life Technologies) plus 10% AB Human Serum (Sigma-Aldrich). Treated cells at the different ratios as shown ($5 \times 10^4$ cells for 2:1, $2.5 \times 10^4$ cells for 4:1 etc.) were washed twice with AIM-V and AB serum medium to eliminate any trace of treatment, suspended in 100 µl AIM-V and AB serum and added to responders, in the presence of 1 µg/ml αCD28 antibody. Three wells were plated per experimental condition. Extra wells with responders alone were plated to monitor cell growth by flow cytometry analysis. Extra wells were stained in subsequent d with Blue LIVE/DEAD Fixable Dead Cell Stain kit (Life Technologies, Cat. #L-23105) in 1× PBS, then cells were diluted in Stain Buffer (FBS) (BD Pharmingen, Cat. #554656) and stained with CD8 PE antibody (TONBO, Cat. #50-0088-T100). After staining, cells were added with 1% formaldehyde in Stain Buffer (FBS) and CFSE peaks of live CD8[+] cells were monitored at LSRII flow cytometer. When optimal growth was achieved all the samples were stained as described above and analyzed on the LSRII.

**Immunoblot studies.** Cells were collected by centrifugation, washed with 1× PBS pH 7.2 and flash-frozen in liquid nitrogen or immediately lysed. Lysis was performed by adding 15 µl ice-cold RIPA buffer per million cells (150 mM NaCl, 50 mM Tris HCl pH 7.5, 1% NP-40, 0.5% Na-deoxycholate, 0.1% SDS, 1 mM EDTA, Thermo Scientific Pearce Protease Inhibitor Tablets, EDTA-free, Thermo Scientific Halt Phosphatase Inhibitor Cocktail) on ice. Cell suspensions were defragmented through a 1 ml syringe several times until no resistance was encountered, and crude extracts were cleared by centrifuging at $20,000 \times g$ 10 min at 4 °C. Protein content for each sample was measured by Pierce BCA Protein Assay Kit (Thermo Scientific). Equal amounts of protein (20–40 µg) were resolved by sodium dodecyl sulphate–polyacrylamide gel electrophoresis and probed by immunoblotting. Membranes were first probed for phospho-proteins, stripped by Restore Western Blot Stripping (Thermo Scientific), then re-probed for total proteins. Primary antibodies were diluted 1:1000 to 1:2000 depending on protein loading. From Cell Signaling: P-STAT5 Y694 (Cat. #9359), STAT5 (Cat. #9358), P-AKT S473 (Cat. #9271), AKT Cat. #9272), P-rpS6 S235/236 (Cat. #2211), rpS6 (Cat. #2217), P-4E-BP1 S65 (Cat. #9456), 4E-BP1 (Cat. #9452), eEF2K (Cat. #3692), eIF4GI (Cat. #2498), eIF4A (Cat. #2013S), GAPDH (Cat. #2118), β-actin (Cat. #4967); from BD: eIF4G2 (Cat. #610742) and eIF4E (Cat. #610270); from BETHYL: eIF3d (Cat. #A301-758A). Secondary ECL[TM] anti-Rabbit IgG or anti-Mouse IgG, horseradish peroxidase linked whole antibody (GE Healthcare, Cat. #NA934V and NA931V, respectively; 1:5000 to 1:10,000). ECL Western Blotting Detection Reagent (GE Healthcare Life Sciences) was used for detection.

**Polysome profile analysis and RNA isolation.** Polysome isolation was performed by separation of ribosome-bound mRNAs via sucrose gradient centrifugation as previously described using 100 µg of polysome RNA from cycloheximide-treated cells, lysed and processed as previously described[65]. Fractions were collected and mRNA purified as described[65]. RNA from total extract and heavy polysomes (four or more ribosomes per mRNA) were extracted by Qiagen RNeasy MiniElute Cleanup kit (Cat. #74204).

**Gene expression array studies and bioinformatics.** In all, 100 ng of RNA per sample was isolated from total cell extracts or from polysome fractions. RNAs were processed using the GeneChip WT PLUS Reagent Kit then hybridized to HuGene 2.0 ST Arrays (Affymetrix, Cat. #902112). Affymetrix chips were processed by the

NYU Genome Technology Center. Gene-level probe set summaries of microarray data were acquired using GCCN and SST transformation algorithms with RMA background correction. Data were quantile normalization using Expression Console Software version 1.4.1 (Affymetrix). We removed from analysis control probe sets and probe sets that lacked mRNA accession tags. Translational efficiency was quantified using the difference in log2 intensity of matched polysomal RNA and total RNA. Differences in transcription and translation were quantified from total RNA and polysome RNA by normalizing each separately. The limma R package was used for further statistical analyses[66].

**Validation of microarray assay by qRT-PCR.** qPCR was performed on RNA used for genome-wide signatures analysis (both transcriptome and translatome). Primers are listed in Supplementary Table 2, and were employed at 200 nM and iTaq Universal SYBR Green Supermix (BioRad). Data were normalized on GAPDH values.

**Pathway analysis of translatome.** Log$_2$ of fold-changes of treated or untreated values were loaded on Ingenuity Pathway Analysis software (Qiagen) and analyzed with a cut off ≥0.5 fold-change difference and a $p$ value ≤ 0.05. The function "Canonical Pathway" was employed and canonical pathways with a significance $-\log(p$ value) ≥2.5 were charted. The z score was also included, which predicts the activation state of the pathway.

**Motif search analysis.** Translatome lists of upregulated (log$_2$ ratio >1) and downregulated mRNAs in RAD001+TGF-beta were used. The 5′-UTR for all the transcripts were extracted from the hg19 Human Reference Genome (UCSC). All the transcript variants for each mRNA were included. Overlapping regions were merged to remove redundant sequences, and UTRs shorter than 50 and longer than 500 bp were excluded. Sequences for most upregulated mRNAs vs. downregulated mRNAs were analyzed by MEME (https://meme-suite.org/meme/doc/meme.html), which found no significant motifs (in single gene set, or vs. background), and GenomeScope (https://academic.oup.com/bioinformatics/article/33/14/2202/3089939), which found enriched motifs. The BNGCNGS motif was highly enriched in the RAD001+TGF-beta upregulated sequences, but it was not discovered in the downregulated list.

**Generation of reporter constructs.** The bicistronic plasmid pRL-HCV IRES-FL[67,68] was employed to generate constructs harboring the 5′-UTR of motif-containing *PRICKLE1* and mTOR-dependent *ITGβ1* mRNAs. 5′-UTRs were amplified from RAD001+TGF-beta-treated cell mRNAs using cDNA reverse transcription cloning. The 5′-UTR of *PRICKLE1* mRNA contained the variant 3 sequence (NCBI Reference Sequence: NM_001144882.1) and the 5′-UTR of *ITGβ1* mRNA contained the variant 1E sequence (NCBI Reference Sequence: NM_133376.2). For the *PRICKLE1_v3* 5′-UTR, restriction site-based cloning was designed using DNAStar tool SeqBuilder, which entailed inserting AflII restriction site at the 5′ end and EcoRV restriction site at the 3′ end, and then digesting and ligating insert and vector with standard procedures. The *ITGβ1* 5′-UTR was inserted by Gibson assembly using NEBuilder HiFi DNA Assembly kit (New England Biolabs, Cat. #E5520) according to manufacturer instructions. Primers were designed via NEBuilder Assembly Tool. The vector was linearized by EcoRV-digestion. Bacteria transformation and colony screening were performed with standard procedure.

**Small interfering RNA (siRNA) silencing.** HEK 293 cells were silenced with 20 nM of siRNAs against DAP5 (ID: s4589), eIF4E (ID: 61117) or non-silencing (Nsi) scramble control (Cat. #4390844), all from Thermo Fisher, using TransIT-siQUEST Transfection Reagent (Mirus, Cat. #MIR2110) following manufacturer instructions. Isolated naïve CD4$^+$ T cells were treated with 1 μM Accell SMARTpool siRNA targeting DAP5 (Cat. #E-011263-00-0020) or a non-targeting Accell control pool (Cat. #D-001910-10-20) from Dharmacon 1 d before activation and on d 3 at 6 after activation. siRNA uptake of a GFP-siRNA non-targeting control (Cat. #D-001950-01-20) was assessed by flow cytometry and the efficiency of silencing was assessed by DAP5 immunoblot.

**Translation assay in transfected HEK 293 cells.** HEK 293 cells were plated at $1.5 \times 10^5$ cells, silenced for DAP5 and eIF4E as described above and the next day transfected with a pRL-HCV IRES-FL plasmid harboring the 5′-UTR of *ITG1B*, *PRICKLE1*, *CD101*, *CD103*, or *FOXP3*. Cells were collected after 72 h in RNase-free conditions and cell number and viability was assessed by trypan blue exclusion in a hemocytometer. Cells were then pelleted, washed with sterile 1× PBS and divided in two. Half of the cells were resuspended in 50 μl Passive Lysis buffer (Promega) per million cells to assess Renilla luciferase activity, while the remaining half was snap frozen and stored at −80 °C for RNA extraction. RNA was extracted by RNeasy Mini kit (Qiagen) with on column DNase treatment by RNase-free DNase Set. RNA was subjected to a second round of DNase digestion by RQ1 RNase-free DNase (Promega) and retrotranscribed via GoScript Reverse Transcription System (Promega). qPCR for Renilla was performed at 200 nM and iTaq Universal SYBR Green Supermix (BioRad). Renilla mRNA quantities were obtained by

interpolating results with a standard curve with serial dilution of pRL-HCV IRES-FL. Luciferase assay results were normalized on Renilla mRNA quantities and protein content measured by BCA Protein Assay (Thermo Fisher Scientific).

**In vitro transcription.** RNAs were generated by in vitro transcription with T7 RNA polymerase (NEB), performed in the presence of 7-methylguanosine cap structure (NEB M0276) or non-functional cap analog (ApppG) (Jena Bioscience, Cat. #NU941) using linearized bicistronic plasmid containing T7 promoter, pRL-HCV IRES-FL as the template harboring the 5′-UTR of ITGβ1, CD101 or CD103 as described above. The capped RNA was polyadenylated using polyA polymerase (NEB, Cat. #M2080S). RNAs were purified by phenol–chloroform extraction and ethanol precipitation.

**In vitro translation.** In vitro translation extracts were made from 293 T cells as we previously described[22]. In brief, lysates were nuclease-treated with 18 gel U/μl micrococcal nuclease (NEB Cat. #M0247S) in the presence of 0.7 mM CaCl$_2$ for 10 min at 25 °C, and the digestion was stopped by addition of 2.24 mM EGTA. Each translation reaction contained 50% in vitro translation lysate (from 293 T cells) and buffer to make the final reaction 0.84 mM ATP, 0.21 mM GTP, 21 mM creatine phosphate (Roche), 45 U/ml creatine phosphokinase (Roche), 10 mM HEPES-KOH, pH 7.6, 2 mM DTT, 8 mM amino acids (Promega), 255 mM spermidine, 1 U/ml murine RNase inhibitor (NEB), and mRNA-specific concentrations of Mg(OAc)$_2$ and KOAc. Translation reactions were incubated for 1.5 h at 30 °C, after which Renilla luciferase activity was assayed.

293 T cell extracts were engineered to inducibly express non-silencing and silencing DAP5 using tetracycline-inducible lentiviral pTRIPZ vector as described[22]. 293 T cell were engineered to silence *EIF3d* at the 3′-UTR, followed by the overexpression of a cap-binding mutant pcDNA5/FRT eIF3d (HA), Helix mutant α5 or Helix α11. The Helix mutant plasmids retain the wild type levels of RNA binding as a part of the eIF3 complex but mutated specifically in its 5′ mRNA cap recognition site as described[62]. In all, 500 ng of in vitro transcribed mRNA was programmed with a cap for in vitro translation. Renilla luciferase activity was measured and normalized to backbone vector. The mutant Helix α5-and Helix α11 vectors were provided by Dr. Amy Lee (Harvard University, Boston, MA).

**Luciferase assay.** Renilla luciferase signal was generated and recorded by employing Dual Luciferase Reported Assay System (Promega) and SpectraMax L Luminometer (Molecular Devices) according to manufacturer instructions.

**Quantification and statistical analysis.** Unpaired $t$ tests and two- or one-way analysis of variance (ANOVA) tests paired or unpaired as indicated were used for biological studies when applicable to determine statistical significance with Dunnett post-ANOVA test determination. Ordinal measurements used Fisher's Exact Test. As necessary, analyses used Benjamin–Hochberg or Tukey's corrections for multiple analyses, sample size. Data were analyzed using GraphPad Prism 9. Minimum significant values are considered $P < 0.05$. Data are expressed as indicated in Figure legends as means with standard error of means, and when appropriate, corrected for sample sizes using Bonferroni corrections to adjust alpha values.

**Reporting summary.** Further information on research design is available in the Nature Research Reporting Summary linked to this article.

## Data availability

The genome-wide data generated in this study have been deposited in the GEO database under accession number GSE178634. All data, including source data, have been provided in the GEO database as indicated above and in Supplementary Data 1–8 found online with this manuscript. All non-commercial reagents including cell lines, plasmids, and other reagents developed in this study are available upon request. Materials described in this paper are available for distribution under the Uniform Biological Material Transfer Agreement, a master agreement that was developed by the NIH to simplify transfers of biological research materials. Source data are provided with this paper.

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

## Acknowledgements

We thank Dr. Juan Lafaille (NYU School of Medicine) for guidance on Treg cell biology, and Stuart Brown (NYU School of Medicine) for the SCOPE analysis. This work was supported by the Breast Cancer Alliance, the Breast Cancer Research Foundation (BCRF-21-146), the Colton Center for Autoimmunity, the US National Institutes of Health (1R01AI37067-01) (to R.J.S.), NIHT32 CA009161 (A.E.), AIRC/Marie Curie International Fellowship from AIRC, the Italian Association for Cancer Research (V.V.), and UL1 TR00038 from the National Center for Advancing Translational Sciences (NCATS) for core services support.

## Author contributions

V.V., S.P.B., C.D.L.P., O.K., and R.J.S. designed the experiments; V.V., S.P.B., C.D.L.P., O.K., and R.J.S. wrote the manuscript; V.V., S.P.B., C.D.L.P., O.K., and S.D. performed the experiments; A.E. performed bioinformatics analyses; V.V, A.E., S.P.B., C.D.L.P., O.K., and R.J.S. analyzed and interpreted the data.

## Competing interests

The authors declare no competing interests.
