## [Peer Review File · Nature Communications]

A DAP5/eIF3d alternate mRNA translation mechanism promotes differentiation and immune suppression function of human regulatory T cellsREVIEWER COMMENTS

Reviewer #1 (Remarks to the Author):

In this article Volta et al, show evidence that during Treg maturation TGFbeta orchestrates transcriptional and translational programs. This encompasses translation of mRNAs encoding factors implicated in differentiation and immunotolerance via DAP5-dependent mechanisms. Notably, this DAP5-dependent translational program is sustained under conditions of low mTOR signaling which is characteristic of Treg maturation. Overall, it was found that this is a well-executed study that addresses an important biological question pertinent to the molecular underpinnings of Treg maturation. Based on this, I thought that this manuscript merits publication and I have some relatively minor concerns and comments that I thought may further improve this study.

Major comments:

-In figure 1 it is surprising that the active site mTOR inhibitor PP242 has a rather modest effects on pS473 on AKT which is mTORC2 phosphoacceptor site, while RAD001 appears to inhibit it at the highest applied concentration. Can the authors comment on this? Is this specific for cell types and/or conditions that were employed in this vs. other studies and/or is this a consequence of toxicity of RAD001 to CD4+ T cells at this concentration? Also, it would be advisable to add an earlier time point to assess acute effects of the drugs on mTOR signaling and employ an additional active-site mTOR inhibitor (e.g. torin1).

-I thought that treatments with RAD001 should not be referred to as "mTORC1 inhibition" considering: i) the length of the treatment whereby in some contexts prolonged rapamycin treatment has also been demonstrated to suppress mTORC2 and ii) rapamycin-insensitive mTORC1 outputs. Ergo, the authors should either specify that these are the effects of RAD001 or use orthogonal genetic approaches (e.g. depletion of raptor) to corroborate that the observed effects are indeed due to selective mTORC1 suppression.

-In figure 3, the authors should consider statistically comparing TGFbeta and TGFbeta+RAD001 conditions. Were these experiments also performed with the active-site mTOR inhibitor?

-Figure 4a should be paralleled by the assay directly measuring global protein synthesis (e.g. S35 labelling, puromycylation etc).

-In figure 4c the levels of eIF4E appear to be higher in TGFbeta treated cells vs. other conditions. Also, TGFbeta alone appears to induce pS65 4E-BP1 signal relative to the control, while combined treatment with TGFbeta and RAD001 appears to suppress pS65 4E-BP1 signal stronger than RAD001 treatment alone. Can the authors comment on this?

-Although widespread in the literature, using log ratios may result in spurious correlations (Pearson 1897) which was indeed shown for normalizing polysome-associated over total mRNA in polysome profiling analysis (Larsson 2010, 2011). Perhaps the authors should consider using a more appropriate analytical method.

-Figure 7f should be supplemented by separately showing RFPRLU and RFP signals and not just the ratio.

-Page 16, lines 3-6: "In contrast, translation of mRNAs controlled by the CD101 5'UTR were reduced by ~8 fold by DAP5 silencing, but only ~3-fold by eIF4E silencing, suggesting greater reliance on DAP5 than eIF4E for translation". Could this also be a consequence of differences in efficacy of depletion of eIF4E vs. DAP5 in these experiment?

-This is mostly my curiosity and perhaps outside of the scope of this paper, but are there effects of sustained or temporal abrogation of DAP5 expression on Treg maturation? Perhaps the authors should comment on this.

Minor concerns:

-Some careful editing was found to be required throughout the manuscript. Some axis labels in the figures also appear to be misaligned.

-The authors should consider italicizing latin terms (e.g. *in vitro*).

-Page 4, lines 17-18: "mTORC1 stimulates translation initiation in part by phosphorylating (inactivating) 4EBP1" Consider changing to 4E-BPs (as far as I know, this is also true for 4E-BP2).

-Page 8, line 23: "With mTORC1 inhibition, most mRNAs should be translationally reduced.." Notwithstanding that mTOR inhibition has profound effects on the translome, translation of subsets of mRNAs are differentially sensitive to mTOR inhibition. To this end, the authors should consider revising the above statement.

-In figure 5a the axis should be changed into "total mRNA abundance" or similar considering that mRNA stability, and not just transcriptional activity, influences total mRNA levels. Same goes for the panel c.

-Page 15, line 10: "We cloned the 5'UTRs of representative Treg mRNAs.." In this sentence, it should be specified which 5'UTRs were used.

-In some figures the number of replicates and types of statistical tests are missing and I think that it would make sense to include them in the figure legends.

I hope that the authors will find my comments constructive and of a sufficient pathos.

Sincerely

I/Topisirovic

Reviewer #2 (Remarks to the Author):

The aim of this manuscript by Volta et al., is to understand an existing paradox concerning the maturation and the immune suppressive activity of peripheral regulatory T cells (Tregs). On the one hand the maturation requires ongoing protein translation, as predicted by its dependence on TGF β signaling, yet on the other hand it depends on continuous mTORC1 inhibition that shuts off the canonical modes of initiation of mRNA translation. To study this enigma the authors used human CD4+ T cells that undergo differentiation in culture by a defined protocol (iTregs). By comparing the genome wide transcriptomic and translomic profiles which develop in response to mTOR inhibition, TGF β or the combination of the two treatments, they identified a set of TGF β -induced mRNAs which continue to be translated under conditions of global translation silencing by mTOR inhibition. Based on transfections with luciferase reporters carrying the 5'UTR of candidate mRNAs, they conclude that the TGF β -induced mRNAs utilize an alternate mechanism of cap-dependent translation that is driven by the DAP5/eIF3d complex. This mechanism is supposed to override the requirement for eIF4G1/eIF4E initiation complex, that is suppressed by mTOR inhibition.

Studying the mechanisms of translation control in T cell maturation/function, is an important research direction, poorly addressed so far. However, there are major concerns in this work which challenge the claim that the translation mechanism has been deciphered. It is not clearly proven that DAP5/eIF3d is part of the translation initiation complex which drives this mode of translation. As detailed below, the luciferase-based translation assays which were planned to study this research direction lack important controls, fail to relate to eIF3d as a major player, and were performed in the wrong cellular context. Overall, the mechanistic basis for this alternate mode of translation remains obscure. Some additional technical and conceptual concerns are detailed below.

Major concerns:

1. The western blots (Figs 1, 4, 7) are of bad quality. A better gel resolution should be presented and the intensity of the relevant bands should be quantified; statistics from repetitive biological assays are required. For example, detailed band quantifications will enable to make a more accurate correlation between the reduction in polysome/monosome ratio by the different treatments (Fig. 4b) and the decline in the phosphorylation of mTOR substrates (Fig. 4c).
2. Fig.3: to support the increased immune suppression by TGF β and RAD001, the characterized hallmarks should be expanded to include additional parameters such as cytokine release assays and membrane staining for Treg markers.
3. The analysis of the transcriptome and the translome, nicely conducted in this manuscript, delineates a group of TGF β -induced mRNA transcripts with translation privilege. It will be beneficial to plan function-based experiments which will prove their necessity for the maturation/function of the Tregs.
4. Fig.7d (not 7c as written in the text by mistake): The data showing that the steady state levels of eIF4G1 and DAP5 proteins change in an opposite direction in response to RAD001 and TGF β (in CD4+ lymphocytes obtained from two donors), do not provide any functional clue about the alternate translation mechanisms.
5. The luciferase reporter assays in Figs 7c and 7f fail to provide a robust proof showing that DAP5/eIF3d initiation complex drives the alternate translation mechanism. The assays are based on comparing the outcome of knocking down of DAP5 or of eIF4E on the expression from reporter vectors carrying a few different 5'UTRs. It is shown that a vector containing the 5'UTR derived from one of the mRNAs belonging to the 'translation privilege' group, displays higher dependence on DAP5 than on eIF4E, while a few other 5'UTRs derived from mRNAs from the other groups were similarly affected by both siRNAs or mostly affected by the eIF4E siRNA. What is clearly missing is a direct proof that eIF3d interacts with DAP5 to form the initiation complex driving the translation of these mRNAs. Experiments referring in one way or another to eIF3d are missing including for example 48S complex isolation studies. Also the authors should prove that the translation of these mRNAs is cap-dependent by replacing the canonical m7GpppG cap to ApppG cap and/or introducing stable hairpin structure (hp) into the vectors. They should exclude a possible IRES activity residing in these 5'UTRs using the canonical bi-cistronic vectors or other strategies available in the field.
6. The luciferase experiments were done in HEK293 cells due to the difficulty to conduct transient transfections or viral infections in CD4+T cells. Notably this is a problematic issue as the different components of the translation machinery may be cell context dependent. Other strategies should be utilized to prove or disprove the proposed concept.
7. Altogether, while the authors rely on their previous publication in which they revealed that DAP5 directly interacts with eIF3d to provide a new mode of translation that is cap-dependent eIF4E-independent (Nat Commun 9, 3068 (2018)) the current information is not sufficient to attribute this mechanism to the Treg system.

Minor concerns

1. Fig. 5c: the comparison between the 3 graphs will be improved if the X and Y axis will be at the same scale.
2. Fig. 7b: the authors should include DAP5 and eIF3d in the blot examining the outcome of cap-chromatography.
3. In Fig.7d, the two left lanes of the phospho 4E-BP1 look problematic (pasted from different exposures/blots?); the loading control in this Fig (GAPDH) shows fluctuations.

Reviewer #3 (Remarks to the Author):

Volta et al. have demonstrated an important role for mTORC1-independent translation in TGF β -induced human iTreg-cells. Specifically, they have shown an alternative mechanism of translation for Treg-associated mRNA transcripts through the initiation factor DAP5.

Some concerns that may be worth addressing are outlined below

1. In Figure 1, Figure 1a suggests a 30% increase in CD25+Foxp3+ cells following RAD001+TGF β treatment. Therefore 70% of the cells within this culture are not iTregs. In addition, the CD25 MFI does not show any difference within the various cultures. However, the finding in the manuscript is interesting and I wonder if the authors can simply state that their data identifies a new signalling pathway driven by TGF-B in T cells rather than iTregs?

2. Would it be possible to show the characterisation of the 70% non-Tregs within this culture of RAD001 and TGF-b? For instance, surface markers, cytokine profile etc? An indepth explanation of their sorting strategy to use CD25 might also clarify their definition of iTregs within this manuscript.

3. In figure 2, only 60% of cells were shown to have Foxp3 expression (Supp Figure 1, Figure 2c) and 51% were CD25+ (up to 78% following positive selection of cells), can the population being used be classified as iTreg cells? Can the authors explain why CD25 was used as a marker for sorting when this is not specifically upregulated in the iTreg culture. The same applies for CD127. Clearly any activated T cell will be CD127 negative and CD25+.

4. In Figure 3b, the summary plots for both 2:1 and 4:1 suppression ratio demonstrate significant and clear suppression in comparison to the untreated control. Would it be possible to show the representative data for these cultures?.

5. In the representative flow plot on suppression, there is no change in the number of peaks between the various control and iTreg cultures. However, most cells in the control culture occupy the most diluted peak. This suggests that the suppression is not robust and a difference in 89.2% vs 71.3% in terms of cell proliferation is of no significance within a biological system. It would be important to show that the iTregs generated actually prevent dilution of effector cells into the same number of peaks as compared to control.

6. Figure 4 is very interesting and exciting with regards to translomics of T-cells upon TGF β stimulation and mTORC1 inhibition. It would be important to analyse the individual polysomal fractions within these conditions; probing for the canonical and non-canonical mRNAs from Figure 7. Upon TGF β stimulation and mTORC1 inhibition, these mRNAs which are dependent on DAP5 and TGF β could then be found in heavier polysome fractions in theory.

7. Similarly, it would be very interesting to show that upon DAP5 silencing by siRNA, the DAP5-dependent mRNAs 'shift' to the lighter polysomal fractions as they are no longer efficiently translated.

8. Furthermore the above approach could be used to show that mTORC1-dependent mRNAs are also up or downregulated following treatments compared to control cultures.

9. Similar to Figure, 4, in figure 6, It would be interesting to validate the mRNA translation differences seen here (Figure 6b and 6c) with a 'shift' in the polysomal fractions they are detected in, to be able to confirm these mRNAs are regulated by TGF β (+/-RAD001).

10. It would also be worth validating the differences in transcription and translation here with a difference in the protein or expression level. This could be done through either immunoblotting or flow cytometry.

11. DAP5 silencing through RNA interference has been shown to induce substantial apoptosis (Marash et al 2008) through cap-independent translation of Bcl-2 and CDK1, with DAP5 playing a role in cell survival as well as stem cell fate/differentiation. With this in mind DAP5-silencing could have multiple off-target effects, feeding into the results seen in Figure 7F. Can the authors discuss this?

12. Figure 7: It would be important to perform RNA interference in primary T cells given the number of working protocols available in the field, including clinical protocols for T cell transduction, in order to further interrogate and validate the role of DAP5 translation in T cells.

Also, polysome profiling and polysomal fraction probing for mRNAs upon DAP5 silencing in T cells would be particularly interesting and confirm the observations in the paper.

13. In Figure 7F, the mTORC1/EIF4E-dependent control of ITG β 1 show over 50% decrease upon DAP5 silencing, thus identifying off-target effects or non-specific effects? Please comment

14. Due to DAP5's role in apoptosis it would be ideal to see cell viability/numbers or apoptosis markers for the cells prior to and following DAP5 silencing.

15. Could Figure 7D include an immunoblot for EIF3d to further show the mechanism of DAP5 translation in these cells. DAP5 has been shown to bind other eIFs and so interrogation of these eIFs would be beneficial.

16. Perhaps EIF3d siRNA silencing could also be used to further identify the specificity of translation for these mRNAs.

17. What is 'siNS' and 'Nsi' in Figure 7C/7F, not specified in the Figure legend. Is it a negative control (No siRNA) or is it Scrambled siRNA? Please clarify

18. It would be beneficial for the study to validate the mRNAs from Figure 6b and 6c in the siRNA Luciferase assay?

19. If possible, please validate lowered translation of the named mRNAs following DAP5 RNA interference, by quantifying protein/expression level after DAP5 silencing.

Minor Concerns

1. Please rewrite figure legends to accurately represent the type of cells used (Fig.4), Mean or Median data shown?, statistical tests (fig.6), specific n numbers (Fig. 2) etc.

2. Please comment on using RAD001 rather than Rapamycin or any other of the many mTORC1 inhibitors?

Reviewer #1

... Overall, it was found that this is a well-executed study that addresses an important biological question pertinent to the molecular underpinnings of Treg maturation. Based on this, I thought that this manuscript merits publication and I have some relatively minor concerns and comments that I thought may further improve this study.

Major comments

1. In figure 1 it is surprising that the active site mTOR inhibitor PP242 has a rather modest effects on pS473 on AKT which is mTORC2 phospho-acceptor site, while RAD001 appears to inhibit it at the highest applied concentration. Can the authors comment on this? Is this specific for cell types and/or conditions that were employed in this vs. other studies and/or is this a consequence of toxicity of RAD001 to CD4+ T cells at this concentration? Also, it would be advisable to add an earlier time point to assess acute effects of the drugs on mTOR signaling and employ an additional active-site mTOR inhibitor (e.g. torin1).

We thank the reviewer for pointing out the need for clarification. Our study is consistent with other studies which found that lymphocytes uniquely response to mTOR inhibitors, which differs dramatically from adherent cell lines (tumor and non-tumor) (see: <https://www.ncbi.nlm.nih.gov/pmc/articles/PMC4924540/>) This is now referenced and noted in our manuscript (page 6, top).

We did not assess acute effects at an earlier time point because naive T cells do not fully differentiate until 4-5 days after IL2 activation, and undergo dramatic reprogramming (even CD4 expression is turned off the first 3 days of activation).

2. I thought that treatments with RAD001 should not be referred to as “mTORC1 inhibition” considering: i) the length of the treatment whereby in some contexts prolonged rapamycin treatment has also been demonstrated to suppress mTORC2 and ii) rapamycin-insensitive mTORC1 outputs. Ergo, the authors should either specify that these are the effects of RAD001 or use orthogonal genetic approaches (e.g., depletion of raptor) to corroborate that the observed effects are indeed due to selective mTORC1 suppression.

While we appreciate the Reviewer's point, as we discussed above, in lymphocytes rapalogs strongly inhibit global translation, 4E-BP phosphorylation and act primarily on mTORC1 not mTORC2. We showed evidence for this as well in Figure 1 using in vitro differentiated naive T cells. We feel that in this case it is appropriate that rapalog treatment can be referred as mTORC1 inhibition.

3. In Figure 3, the authors should consider statistically comparing TGFbeta and TGFbeta+RAD001 conditions. Were these experiments also performed with the active-site mTOR inhibitor?

This comparison is found in new Figure 2g. We now provide statistical analysis of results in new Figure 2g for TGFβ alone compared to TGFβ plus RAD001 (they are statistically significantly different in all but the 2:1 titration). The suppression assay was not performed with active-site mTOR inhibition because as we showed in Figure 1c, dual mTORC1/2 inhibition prevents CD4 T cell proliferation which itself blocks differentiation of T cells. This is now noted on page 7, top.

4. Figure 4a should be paralleled by the assay directly measuring global protein synthesis (e.g., S35 labelling, puromycylation etc).

Now found in new Figure 4c. This is an important point and we thank the reviewer for pointing this out. We now provide in vivo labeling as requested and quantify the reduced level of protein synthesis. This has been added as new Figure 4 panel c. We measured protein synthesis activity by Click-iT Plus OPP fluorescent puromycylation analysis of naive CD4 T cells and those differentiated into Tregs in the

presence of TGFβ and/or RAD001. The reduction in polysomal mRNAs by ~70% corresponds to a decrease in global protein synthesis in RAD001+TGFβ-treated cells by OPP assay of ~65%.

5. In figure 4c the levels of eIF4E appear to be higher in TGFbeta treated cells vs. other conditions. Also, TGFbeta alone appears to induce pS65 4E-BP1 signal relative to the control, while combined treatment with TGFbeta and RAD001 appears to suppress pS65 4E-BP1 signal stronger than RAD001 treatment alone. Can the authors comment on this?

Now new Figure 5d. We thank the reviewer for pointing this out. The variation in eIF4E levels and pS65 4E-BP1 in TGFβ treated cells is not consistent across donors, of which we analyzed >70 PBMC donors. However, the double treatment always results in the strongest suppression in 4E-BP1 phosphorylation. The reviewer's concern prompted us to change the way in which we present these data. Now we normalize across human donor specimens to reduce donor to donor variation, and to obtain enough cells for immunoblot analysis and reduce donor variation, by pooling 4 donor specimens per study rather than using a single representative donor. This is now noted in the text and Methods. This provides more representative data and given the low number of CD4+ T cells obtained per donor, makes it possible to carry out multiple immunoblots. The new immunoblots are in Figure 4d and have also been quantified.

6. Although widespread in the literature, using log2 ratios may result in spurious correlations (Pearson 1897) which was indeed shown for normalizing polysome-associated over total mRNA in polysome profiling analysis (Larsson 2010, 2011). Perhaps the authors should consider using a more appropriate analytical method.

The reviewer is correct that there are other approaches for genome-wide data analysis that can sometimes perform better than the log2 ratio method. However, the small numbers of cells obtained from human samples makes it impossible to use these other approaches, such as Anoto, which require typically up to 5 replicates of the same specimen to generate analyzable data. Human donor PBMC specimens used to isolate the small number of CD4+ T cells are only able to derive enough cells for two replicates per donor. Consequently, we used multiple independent replicates from multiple human donor specimens each repeated twice to provide statistical power, which for the log2 ratio approach, provides accurate and gold-standard data and is more forgiving in its requirement for multiple replicates. We note that we also verified some of the targets identified in our genome-wide study using qRT-PCR and found similar results (see Suppl Fig. 3).

7. Figure 7f should be supplemented by separately showing RFPRLU and RFP signals and not just the ratio.

Because expression levels differ widely among the different mRNAs, in all studies we normalized to control to allow comparisons. For this reason we prefer to not present the raw data in the form mentioned by the reviewer.

8. Page 16, lines 3-6: "In contrast, translation of mRNAs controlled by the CD101 5'UTR were reduced by ~8 fold by DAP5 silencing, but only ~3-fold by eIF4E silencing, suggesting greater reliance on DAP5 than eIF4E for translation". Could this also be a consequence of differences in efficacy of depletion of eIF4E vs. DAP5 in these experiments?

We appreciate the reviewer's point. That is why we were careful to make certain that DAP5 and eIF4E were silenced to similar levels based on immunoblotting in these studies, although the endogenous levels of eIF4E and DAP5 may be different. Nevertheless, the important point in these studies is that mRNAs identified on a genome-wide level that were particularly translationally sensitive to silencing of eIF4E or DAP5, demonstrated sensitivities that reflected their in vivo activities using just their 5'UTRs in a reporter assay.

9. This is mostly my curiosity and perhaps outside of the scope of this paper, but are there effects of sustained or temporal abrogation of DAP5 expression on Treg maturation? Perhaps the authors should comment on this.

Please see our extended response to Reviewer #3 comment 12. We have now added an additional new figure (Figure 8) in which we developed an approach for silencing DAP5 in human naïve CD4+ T cells (which has rarely been achieved) and show that it significantly impairs their ability to differentiate into Tregs in culture. We also conducted the experiment Reviewer suggested, by attempting to silence DAP5 in fully differentiated Tregs. Unfortunately, differentiated Tregs could not be silenced by every method tested.

Minor concerns

1. Some careful editing was found to be required throughout the manuscript. Some axis labels in the figures also appear to be misaligned.

This has all been corrected.

2. The authors should consider italicizing Latin terms (e.g., *in vitro*).

This has been done.

3. Page 4, lines 17-18: “mTORC1 stimulates translation initiation in part by phosphorylating (inactivating) 4EBP1” Consider changing to 4E-BPs (as far as I know, this is also true for 4E-BP2).

This has been corrected.

4. Page 8, line 23: “With mTORC1 inhibition, most mRNAs should be translationally reduced..” Notwithstanding that mTOR inhibition has profound effects on the translome, translation of subsets of mRNAs are differentially sensitive to mTOR inhibition. To this end, the authors should consider revising the above statement.

This has been corrected.

5. In figure 5a the axis should be changed into “total mRNA abundance” or similar considering that mRNA stability, and not just transcriptional activity, influences total mRNA levels. Same goes for the panel c.

This has been corrected.

6. Page 15, line 10: “We cloned the 5’UTRs of representative Treg mRNAs..” In this sentence, it should be specified which 5’UTRs were used.

This has been corrected.

7. In some figures the number of replicates and types of statistical tests are missing and I think that it would make sense to include them in the figure legends.

We thank the reviewer for pointing this out. This has been corrected.

Reviewer #2 (Remarks to the Author):

... Studying the mechanisms of translation control in T cell maturation/function, is an important research direction, poorly addressed so far. However, there are major concerns in this work which challenge the claim that the translation mechanism has been deciphered. It is not clearly proven that DAP5/eIF3d is part of the translation initiation complex which drives this mode of translation. As detailed below, the luciferase-based translation assays which were planned to study this research direction lack important controls, fail to relate to eIF3d as a major player, and were performed in the wrong cellular context. Overall, the mechanistic basis for this alternate mode of translation remains obscure. Some additional technical and conceptual concerns are detailed below.

Major concerns:

1. The western blots (Figs 1, 4, 7) are of bad quality. A better gel resolution should be presented and the intensity of the relevant bands should be quantified; statistics from repetitive biological assays are required. For example, detailed band quantifications will enable to make a more accurate correlation between the reduction in polysome/monosome ratio by the different treatments (Fig. 4b) and the decline in the phosphorylation of mTOR substrates (Fig. 4c).

We apologize for the poor image resolution of some of the immunoblots. It was not so much a matter of poor gel and immunoblot quality as it was the need to enlarge the images in order to provide better resolution. In some cases, we have replaced immunoblots with others that have better resolution. As for quantification, in cases where gels were shown simply to represent large qualitative differences with and without silencing, for instance, there is no need to quantify. For example, the results of Figure 1a or levels of silencing that don't require quantification as in Figures 7b, e and g. In all other cases we have now provided quantification of bands, shown below each set of data. Regarding quantitation of protein synthesis activity, as described for Reviewer 1 (comment #4), we now provide in vivo translation activity data.

2. Fig.3: to support the increased immune suppression by TGF β and RAD001, the characterized hallmarks should be expanded to include additional parameters such as cytokine release assays and membrane staining for Treg markers.

We agree that showing additional markers would further enrich our findings. Due to Treg heterogeneity and plasticity, Treg immunosuppressive activity remains the most reliable hallmark for their characterization as regulatory immune suppressive cells which we provided (now found in new Figure 2g). Nonetheless, we assessed by flow cytometry additional Treg membrane markers, such as the highly specific Treg marker GITR, a key marker of functional Tregs (PMID: 25961057, 11869690, 30427630), and canonical Treg markers such as CD101, CD103, and Treg-specific immunomodulatory cytokines TGF β and IL-10. CD4⁺CD127^{dim}-CD25⁺Foxp3⁺ iTregs were found to concomitantly express high levels of GITR, CD101, CD103, TGF β and IL-10. The percentage of CD101^{hi} and CD103^{hi} cells within the Treg compartment (CD4⁺CD127^{dim}-CD25⁺Foxp3⁺ cells), also increased after the combined treatment (RAD001 +TGF β) that produces the highest level of Treg immune suppression activity. We have now included these additional supporting results as new Figure 3. Original Figure 3 is now presented as new Figure 2a and new Suppl Figure 2a.

3. The analysis of the transcriptome and the translome, nicely conducted in this manuscript, delineates a group of TGF β -induced mRNA transcripts with translation privilege. It will be beneficial to plan function-based experiments which will prove their necessity for the maturation/function of the Tregs.

We understand the reviewer's point, but note that numerous published studies previously established that these genes are required collectively for Treg development and function (which we referenced). We ask the reviewer's understanding that the purpose of these studies was not to identify the importance of these Treg mRNAs in Treg development which is well established, but rather, the importance of DAP5/eIF3d in their translation, which we believe we have done convincingly.

4. Fig.7d (not 7c as written in the text by mistake): The data showing that the steady state levels of eIF4G1 and DAP5 proteins change in an opposite direction in response to RAD001 and TGF β (in CD4⁺ lymphocytes obtained from two donors), do not provide any functional clue about the alternate translation mechanisms.

We thank the reviewer for raising this important issue to address to improve our study. This issue overlaps with issue #5 raised by this Reviewer- both are now fully addressed below by additional experiments that improve our study. We appreciate the reviewer's concern regarding the need for more data regarding functional evidence for DAP5/eIF3d in the translation of Treg lineage determining mRNAs. We now provide additional studies as requested (described below in #5), which we believe compellingly demonstrates the primary importance of both DAP5 and eIF3d in Treg cap-dependent mRNA translation.

5. The luciferase reporter assays in Figs 7c and 7f fail to provide a robust proof showing that DAP5/eIF3d initiation complex drives the alternate translation mechanism. The assays are based on comparing the outcome of knocking down of DAP5 or eIF4E on the expression from reporter vectors carrying a few different 5'UTRs. ... What is clearly missing is a direct proof that eIF3d interacts with DAP5 to form the initiation complex driving the translation of these mRNAs. Experiments referring in one way or another to eIF3d are missing including for example 48S complex isolation studies. Also, the authors should prove that the translation of these mRNAs is cap-dependent by replacing the canonical m7GpppG cap to ApppG cap and/or introducing stable hairpin structure (hp) into the vectors. They should exclude a possible IRES activity residing in these 5'UTRs using the canonical bi-cistronic vectors or other strategies available in the field.

We thank the reviewer for suggesting additional studies which we have now completed that further supports our conclusions. We previously demonstrated that DAP5 and eIF3d directly interact, shown by high resolution mass spectrometry of protein-protein complexes and by direct in vivo (live cell) cross-linking between the two proteins (de la Parra et al., Nature Comm. 2018). We also previously showed that both DAP5 and eIF3d are required to form 48S pre-initiation complexes on DAP5/eIF3d-dependent mRNAs. To address the Reviewer's concern, we now provide additional evidence that DAP5, eIF3d cap binding activity and a cap on mRNA are all required for DAP5-eIF3d directed mRNA translation of Treg fate-determining mRNAs. We first provide immunoblots from naïve CD4+ T cells and induced Tregs demonstrating that TGF β + RAD001 induction of Treg differentiation involves strong upregulation of DAP5 and eIF3d, and downregulation of eIF4G1 expression (New Figure 7a). We then carried out cap-chromatography using 293 cells silenced for either eIF4E, eIF3d or control silenced. We show for the first time eIF3d retention by m7G-Sepharose beads, which was increased with eIF4E silencing (new Figure 7b). Identification of the protein as eIF3d that is retained by cap-chromatography was shown to be correct because it was eliminated by its silencing.

We then developed in vitro translation extracts from wild type, DAP5 silenced and eIF3d silenced 293 cells, and programmed them with luciferase reporter mRNAs containing the 5'UTR of control integrin β 1 (ITG β 1) mRNA which is strongly eIF4E-dependent, or Treg mRNAs that are strongly DAP5/eIF3d dependent (CD101, CD103). To determine whether translation of the DAP5/eIF3d-dependent Treg mRNAs are cap-dependent, we carried out in vitro translation without and with the inhibitory cap-analog, ApppG. Treatment with the ApppG cap analog inhibited translation of all test mRNAs: ITG β 1, Treg CD101 and CD103 mRNAs, demonstrating that the translation of all of these mRNAs is cap-dependent, whether mediated by eIF4E or eIF3d cap binding (new Figure 7c). Next, we silenced DAP5 and repeated the in vitro translation studies (new Figure 7d). DAP5 silencing did not impair translation directed by the ITG β 1 5'UTR, but strongly reduced CD101 and CD105 5'UTR directed translation, showing that both DAP5 and eIF3d are required for their translation. We then show that the cap-binding activity of eIF3d is required for its ability to promote translation of Treg CD101 and CD105 mRNAs but has no effect on translation directed by the ITG β 1 5'UTR (new Figure 7f). Finally, we silenced wild type eIF3d in siRNA transfected 293 cells and replaced it with mutant eIF3d that lacks cap-binding activity (kindly provided by Dr. Amy Lee, Harvard University). Both the α 5 and α 11 mutant eIF3d proteins that lack cap-binding activity strongly reduced translation of luciferase reporter mRNAs containing the 5'UTRs of CD101 and CD105, but not that of the reporter containing the ITG β 1 5'UTR. We believe that collectively, these additional new data greatly improve our manuscript and we thank the reviewer for raising this issue.

Regarding the question of cellular context raised by the reviewer, we have now shown cap and DAP5/eIF3d selective translation directed by the 5'UTR of Treg DAP5/eIF3d-dependent mRNAs in 293 cells. We therefore believe that with these new data addresses the concern about cell context raised by the reviewer has been addressed, which has improved our manuscript and there is no need for additional studies in yet another cell type (Jurkat cells). Moreover, we were not able to develop efficient in vitro translation extracts from Jurkat cells.

6. The luciferase experiments were done in HEK293 cells due to the difficulty to conduct transient transfections or viral infections in CD4+T cells. Notably this is a problematic issue as the different components of the translation machinery may be cell context dependent. Other strategies should be utilized to prove or disprove the proposed concept.

We thank the reviewer for raising this issue. We believe we have addressed the cell context concern as described above, and with the inclusion of DAP5 silencing of human CD4+ T cells in new Figure 8.

7. Altogether, while the authors rely on their previous publication in which they revealed that DAP5 directly interacts with eIF3d to provide a new mode of translation that is cap-dependent eIF4E independent (Nat Commun 9, 3068 (2018)) the current information is not sufficient to attribute this mechanism to the Treg system.

Please see response to criticism # 5, above. We believe this important concern has now been addressed. Regarding translation of Treg mRNAs, eIF3d silencing of CD4+ T cells does not prove specificity for Treg differentiation because it is required for eIF3 function and it is not possible to both silence it in CD4+ T cells and stably transfect these cells with an expression vector for the cap-mutant form of eIF3d. T cells are notoriously difficult to transfect or infect with lentivirus vectors, and combined with the 13 days required to differentiate them in culture makes this an impossible experiment. We believe that with our new data provided in new Figure 7 showing the requirement for both DAP5 and cap-binding by eIF3d in translation of essential Treg differentiation mRNAs, and the essential requirement for DAP5 in Treg mRNA translation and Treg development, that the point has been well proven.

Minor concerns

1. Fig. 5c: the comparison between the 3 graphs will be improved if the X and Y axis will be at the same scale.

We ask that the reviewer please note that the RAD001 data is much less robust than the TGF β data, and therefore cannot be scaled with the same values, which condenses the RAD001 data to the point that they cannot be visualized. We now make note in the text that the data are scaled differently for this reason.

2. Fig. 7b: the authors should include DAP5 and eIF3d in the blot examining the outcome of cap chromatography.

This has now been included in new Figure 7b as described above in response to the reviewer's comment. #5.

3. In Fig.7d, the two left lanes of the phospho 4E-BP1 look problematic (pasted from different exposures/blots?); the loading control in this Fig (GAPDH) shows fluctuations.

These data are all obtained from the same study and were not a product of splicing lanes. However, as explained in response to Reviewer 1 comment #5, we have now changed the way in which protein data are presented. As a result, many of the immunoblots have been replaced and we made certain to improve their quality, although the results are the same as previously shown. The new data in this case is in new Figure 7a.

Reviewer #3 (Remarks to the Author):

Volta et al. have demonstrated an important role for mTORC1-independent translation in TGF β -induced human iTreg-cells. Specifically, they have shown an alternative mechanism of translation for Treg associated mRNA transcripts through the initiation factor DAP5.

Some concerns that may be worth addressing are outlined below.

1. In Figure 1, Figure 1b suggests a 30% increase in CD25+Foxp3+ cells following RAD001+TGF β treatment. Therefore 70% of the cells within this culture are not iTregs. In addition, the CD25 MFI does

not show any difference within the various cultures. However, the finding in the manuscript is interesting and I wonder if the authors can simply state that their data identifies a new signaling pathway driven by TGF-B in T cells rather than iTregs?

Our apology for the confusing presentation of these data. This has now been clarified in the revised text. The remaining cells are uncommitted CD4+ T cells which is why they were not further analyzed. It is typical when purified from PBMCs to have a large fraction of CD4+ cells that cannot be induced to differentiate into different lineages. We now make clear that the percentage of cells refers to those that were isolated from the total population of CD4+ cells which express well accepted, conventional Treg markers. All subsequent studies, including genome-wide analyses, then used the highly purified Treg population, not the total CD4+ cell population which I believe was the Reviewer's concern. We now make this clear in the text. With respect to the CD25 MFI, CD25 does in fact increase in TGF β , RAD001 and RAD001+TGF β treated samples as expected, since all of the cells were isolated using Treg-specific markers and have Treg suppressing activity, though endowed with different immune suppression activities.

2. Would it be possible to show the characterization of the 70% non-Tregs within this culture of RAD001 and TGF-b? For instance, surface markers, cytokine profile etc? An in-depth explanation of their sorting strategy to use CD25 might also clarify their definition of iTregs within this manuscript.

We thank the reviewer for identifying the need for greater clarity in our discussion of these studies. The sorting strategy is now detailed more clearly in Methods and the legend to Figure 2 and Suppl Fig 1. More than 60% of cells express FoxP3, depending on the donor and sample. Of note, in peripheral blood, the Treg percentage is 0.7% - 1.7% of circulating leukocytes. As noted above, the peripheral non-Tregs in the CD4+ isolated population of cells from PBMCs represent CD4+ T cells that did not commit to differentiate to Tregs. These cells were purified away from the Treg population and not further analyzed – they represent a mixture of other CD4 T cells including uncommitted naïve T cells. The differentiated Treg cells were further isolated using standard procedures as described in the text and comprise a highly enriched population that were used for all studies.

3. In figure 2, only 60% of cells were shown to have Foxp3 expression (Supp Figure 1, Figure 2c) and 51% were CD25+ (up to 78% following positive selection of cells), can the population being used be classified as iTreg cells? Can the authors explain why CD25 was used as a marker for sorting when this is not specifically upregulated in the iTreg culture. The same applies for CD127. Clearly any activated T cell will be CD127 negative and CD25+.

We thank the reviewer for noting the need for further clarification of this point. Tregs were sorted and isolated from the CD4+ population using these established markers not just CD25+: CD4+, CD25+, CD127- cells. This is the standard and well accepted procedure for isolation of human CD4+ Treg cells that then results in highly enriched immune suppressing CD4 Tregs. The suppression assay then verified that the cells were indeed Tregs, capable of inhibiting responders' proliferation. We have revised the text accordingly to make this clear.

4. In Figure 3b, the summary plots for both 2:1 and 4:1 suppression ratio demonstrate significant and clear suppression in comparison to the untreated control. Would it be possible to show the representative data for these cultures?

Former Figure 3b is now new Figure 2g and new Suppl Figure 2a. We did not provide these raw data because suppression ratios at 2:1 and 4:1 are not considered a good indication of specific Treg suppression activity, as the more numerous cells could be decreasing proliferation of responders by consuming nutrients. We therefore provided histograms only for the data corresponding to the higher suppression ratios.

5. In the representative flow plot on suppression, there is no change in the number of peaks between the various control and iTreg cultures. However, most cells in the control culture occupy the most diluted

peak. This suggests that the suppression is not robust and a difference in 89.2% vs 71.3% in terms of cell proliferation is of no significance within a biological system. It would be important to show that the iTregs generated actually prevent dilution of effector cells into the same number of peaks as compared to control.

Our apology if this was not clear. We have now more clearly explained in the analysis of cell distributions in the suppression assay studies (Immune Suppression section in Methods). The change in distribution of the cells across the peaks is the measure of suppression. The activated responders are mostly in the left peak, meaning most cells have divided multiple times. In the treatments, the right peak is higher, indicating many cells had not divided at all. Depending on the treatment, the cells divide more or less and distribute themselves in the different intermediate peaks. Calculations were made as described in the text to capture the differences, and the resulting percentage of suppression plotted in the histograms. The numbers the reviewer refers to are therefore not the appropriate quantitation of suppression activity.

6. Figure 4 is very interesting and exciting with regards to translationalomics of T-cells upon TGF β stimulation and mTORC1 inhibition. It would be important to analyze the individual polysomal fractions within these conditions; probing for the canonical and non-canonical mRNAs from Figure 7. Upon TGF β stimulation and mTORC1 inhibition, these mRNAs which are dependent on DAP5 and TGF β could then be found in heavier polysome fractions in theory.

We now more clearly explain these studies in the text. Regarding the Reviewer's interest in detecting subtle changes in translation by ribosome content on mRNA, unfortunately it is beyond the capability of present technologies. It is not reproducible to compare mRNA levels between individual fractions of polysomes, for instance comparing 3 ribosome to 4 or 5 ribosome fractions because of biological flux and sample heterogeneity are too large. As a result, translationalomics analysis was performed by pooling the moderate and well translated polysome fractions (≥ 4 ribosomes), which represent the majority of translation activity, which are then compared to total mRNA levels as log₂ ratios. Suppl Figure 5 shows the behavior of individual mRNAs within the moderate-well translated fraction compared to total mRNA levels with different treatment conditions.

7. Similarly, it would be very interesting to show that upon DAP5 silencing by siRNA, the DAP5-dependent mRNAs 'shift' to the lighter polysomal fractions as they are no longer efficiently translated.

Please see the response to comment #6 above, which is a similar question regarding detection of subtle changes in mRNA ribosome content.

8. Furthermore the above approach could be used to show that mTORC1-dependent mRNAs are also up or downregulated following treatments compared to control cultures.

We apologize if this was not clear. We provide these data in Figure 5a and 5c, in the scatter plot analyses. The raw ranked data for these analyses is provided as well, where statistically significant changes of ribosome-mRNA content are generally 1.5-fold and above.

9. Similar to Figure, 4, in figure 6, It would be interesting to validate the mRNA translation differences seen here (Figure 6b and 6c) with a 'shift' in the polysomal fractions they are detected in, to be able to confirm these mRNAs are regulated by TGF β (+/-RAD001).

Again, apologies if this was not clear in the original text which has been clarified. As noted above, detecting reproducible subtle changes in ribosome loading on mRNAs is beyond the capability of present technologies, whether polysome or ribosome footprint analyses, which can statistically detect only larger changes in a reproducible manner. This is even more difficult when using donor specimens like PBMCs where there is inherent variation from donor to donor. Nevertheless, as shown in Figure 6, we do detect and identify significant large changes in both translation and transcription (mRNA abundance). Regarding how well changes in ribosome content on mRNA reflects protein expression, we provide additional data demonstrating that ~70% reduction in overall ribosome content on mRNAs corresponds to an ~65%

reduction in *in vivo* mRNA translation activity (new Figure 4b, c). We also provide additional data for Treg biomarker expression by flow cytometry in untreated and DAP5 silenced cells.

10. It would also be worth validating the differences in transcription and translation here with a difference in the protein or expression level. This could be done through either immunoblotting or flow cytometry.

We thank the Reviewer for raising this important point. We agree with the reviewer that validating some of the differences in how mRNA-polysome content relates to protein levels will further enrich our findings. As suggested, in new Figure 3, we have now assessed by flow cytometry canonical Treg markers found to be reprogramed in our genome-wide data: CD25, Foxp3 (both shown previously), and now also CD101, CD103, GITR, IL-10 and TGF β . The percentage of Foxp3, CD25, CD101^{hi}, CD103^{hi} GITR, IL-10 and TGF β CD4+ T cells within the Treg compartment increased after the combined treatment of RAD001+TGF β , compared to treatment of RAD001 or TGF β alone or untreated cells. These data substantiate that loss of protein correlates with the loss of mRNAs from moderate-well translated polysomes.

11. DAP5 silencing through RNA interference has been shown to induce substantial apoptosis (Marash et al 2008) through cap-independent translation of Bcl-2 and CDK1, with DAP5 playing a role in cell survival as well as stem cell fate/differentiation. With this in mind DAP5-silencing could have multiple off-target effects, feeding into the results seen in Figure 7F. Can the authors discuss this?

The reviewer has raised an important point that we now clarify. In the Marash study, expression of DAP5 after silencing was extremely low or practically undetectable. In our studies, in contrast, we silenced DAP5 protein levels by no more than ~70% and consequently we did not observe increased cell death with silencing. We now provide both CD4+ T cell and 293 cell viability data without and with DAP5 silencing, and show that in non-differentiating CD4+ T cells there was no increased cell death with DAP5 silencing (new Figure 8) which was also not observed in 293 cells (new Suppl Figure 5d).

12. Figure 7: It would be important to perform RNA interference in primary T cells given the number of working protocols available in the field, including clinical protocols for T cell transduction, in order to further interrogate and validate the role of DAP5 translation in T cells. Also, polysome profiling and polysomal fraction probing for mRNAs upon DAP5 silencing in T cells would be particularly interesting and confirm the observations in the paper.

The reviewer raises an important point, and one that is technically quite challenging for Tregs which are highly resistant to transfection and transduction. We have however managed to carry out this study which required quite some time, optimization and expense (Accell siRNAs are very expensive). We hope that our approach will be useful to others who have also struggled to silence gene in Tregs. We used a pool of cholesterol-linked siRNAs to DAP5 known as Accell siRNAs and developed culture conditions that allowed repeated (every 3 day) transfection of CD4+ T cells in 0% serum during transfection over a period of 2 weeks required for their differentiation with excellent viability (new Figure 8). We achieved an ~3-fold reduction in DAP5 protein levels in activated CD4+ T cells induced to differentiate with TGF β + RAD001, which resulted in a 60% reduction in development of Tregs. This was quantified by flow cytometry using Treg specific markers including highly specific GITR (CD25⁺ CD127^{dim}- Foxp3⁺ GITR⁺ population). We also quantified the viability of CD4+ T cells after DAP5 silencing and observed a similar 60% reduction, but high viability in the non-differentiated CD4+ T cell compartment. 0% serum is essential for transfection because serum cholesterol competes with siRNA uptake. As noted in response to Reviewer 1 comment #9, silencing of DAP5 in fully differentiated Treg cells was not possible. However, our data do allow us to firmly conclude that DAP5 is required for new Treg development. We could not test reversion of differentiated Tregs by DAP5 silencing because we could never silence them using every approach and technology that is available. This is now developed in the Discussion section of the manuscript. We could not achieve greater than 60-70% reduction in DAP5 protein expression, but these data we believe are quite compelling and prove the importance of DAP5 in Treg development.

13. In Figure 7F, the mTORC1/EIF4E-dependent control of ITG β 1 shows over 50% decrease upon DAP5 silencing, thus identifying off-target effects or non-specific effects? Please comment.

We now clarify the text to address this question (now new Figure 7). We do not believe this is an off-target effect, but as we show, instead indicates that ITG β 1 can be translated at a much low efficiency by the DAP5/eIF3d complex, but has a much stronger dependence on the canonical eIF4E/eIF4G complex. ITG β 1 was reduced ~15-fold by eIF4E silencing compared to 2-fold for DAP5/eIF3d silencing.

14. Due to DAP5's role in apoptosis it would be ideal to see cell viability/numbers or apoptosis markers for the cells prior to and following DAP5 silencing.

We thank the reviewer for noting this point which we also addressed with respect to Reviewer # 3's comment #11 (above). We found no reduction in 293 cell viability (new Suppl Figure 5d) and no reduction in naïve CD4+ T cell viability prior to TGF β + RAD001 induction of differentiation (new Figure 8). non-differentiating CD4+ T cell viability with silencing of DAP5 or eIF3d at the levels of silencing used.

15. Could Figure 7D include an immunoblot for EIF3d to further show the mechanism of DAP5 translation in these cells. DAP5 has been shown to bind other eIFs and so interrogation of these eIFs would be beneficial.

This has now been provided, including for eIF3d which was not previously included. We found from multiple donors that with CD4+ Treg development, levels of eIF3d increased only in the combined RAD001 + TGF β treated specimens (new Figure 7a). We also now provide data pooled from 4 independent donor CD4 T cells, which is more representative and we expanded the analysis to additional factors as requested, none of which were found to change with CD4 differentiation except eIF4G1, DAP5 and eIF3d.

16. Perhaps eIF3d siRNA silencing could also be used to further identify the specificity of translation for these mRNAs.

We have now expanded these studies, as described in response to Reviewer #1 comment #9, as discussed above.

17. What is 'siNS' and 'Nsi' in Figure 7C/7F, not specified in the Figure legend. Is it a negative control (No siRNA) or is it Scrambled siRNA? Please clarify.

We apologize for the typo (Nsi) in Figure 7f. siNS is a scrambled non-silencing siRNA. We have now corrected the typo in the figure and in the legend.

18. It would be beneficial for the study to validate the mRNAs from Figure 6b and 6c in the siRNA Luciferase assay?

Three canonical (CD101, CD103, FOXP3) and one non-canonical (PRICKLE1) mRNA listed in Figure 6 were used for the siRNA luciferase assay shown in new Figure 7.

19. If possible, please validate lowered translation of the named mRNAs following DAP5 RNA interference, by quantifying protein/expression level after DAP5 silencing.

We apologize if the text was not clear. Figure 7c examined the translation of luciferase reporter mRNAs containing the different 5'UTRs. Luciferase activity is therefore the product of luciferase protein levels directed by the different 5'UTRs.

Minor Concerns

1. Please rewrite figure legends to accurately represent the type of cells used (Fig.4).

This has now been corrected.

Mean or Median data shown?

MFI is Median Fluorescence Intensity. This is now more clearly indicated.

Statistical tests (fig.6)

P-values from analyses are indicated along with the types of statistical analyses used in all legends.

Specific n numbers (Fig. 2)

Number or repetitions are indicated in the legend.

2. Please comment on using RAD001 rather than Rapamycin or any other of the many mTORC1 inhibitors?

Rapamycin's clinical derivative which has better solubility and potency, RAD001/everolimus, is the preferred drug of choice.

REVIEWER COMMENTS

Reviewer #1 (Remarks to the Author):

I thought that the authors have provided significant new data that further strengthened their conclusions and that they provided satisfactory comments to all my comments. To this end, I have no further concerns regarding this manuscript.

Sincerely

I/Topisirovic

Reviewer #2 (Remarks to the Author):

The reviewer is satisfied with the revised manuscript. The new data improved the status of the manuscript. All the comments were successfully addressed. We understand that for running the translation experiments, a switch to HEK 293 cell lysates was practically essential. Despite the deviation from the main experimental system which may be problematic, these cell free system data are important and improve the paper.

Minor comments:

In Fig 7h- the control bars should be in black color.

Fig.4e- is mentioned in the text and does not appear – the text refer to Fig.4d

Reviewer #3 (Remarks to the Author):

Please address queries addressing experimental data or representative data in points 2, 3, 4. Show representative flow data for TGF-b, IL-10 and G1TR in point 10.

point 5, even if we follow the authors interpretation of the suppression data, that, it is the number of cells within each peak which denotes suppression and not the number of divided peaks (although in my opinion it is usually both!), if one looks at the cultures with TGF-b and TGF-b+RAD001, there doesn't seem to be a significant difference. In addition, the counts (y axis) are not similar in all the cohorts, hence the change in distribution of cells although may show suppressive potential, the data with RAD001 plus TGF-b is not biologically significant as compared to TGF-b alone.

Point 11- do the authors show viability test in 5d? or 6c? It would be important to perform viability test with Annexin V/ 7AAD analysis in order to determine apoptosis that occurs with RNA interference assays. Also, in Figure 8, I do not see any viability data.

Minor concern, expand on whether the error bars in the column graphs are a representation of SEM or SD not the MFI.

Response to reviewer comments (reviewer comments in black type, author response in blue type)

Reviewer #1 (Remarks to the Author):

I thought that the authors have provided significant new data that further strengthened their conclusions and that they provided satisfactory comments to all my comments. To this end, I have no further concerns regarding this manuscript.

Reviewer #2 (Remarks to the Author):

The reviewer is satisfied with the revised manuscript. The new data improved the status of the manuscript. All the comments were successfully addressed. We understand that for running the translation experiments, a switch to HEK 293 cell lysates was practically essential. Despite the deviation from the main experimental system which may be problematic, these cell free system data are important and improve the paper.

Minor comments:

In Fig 7h- the control bars should be in black color.

This has been corrected.

Fig.4e- is mentioned in the text ... refer to Fig.4d.

This has been corrected.

Reviewer #3 (Remarks to the Author):

Please address queries addressing experimental data or representative data in points 2, 3, 4. Show representative flow data for TGF-b, IL-10 and GITR in point 10.

Please note that we did previously respond to queries 2, 3 and 4 – our apology if the Reviewer missed our responses. Nonetheless, we now elaborate further in these responses, both in the text and here as well.

Regarding queries 2 and 3, as we explained in the previous Response to Reviewers and now expanded upon here and in the text (see pages 8-9), Tregs were analyzed and isolated from the CD4⁺ population using standard, broadly established markers for Treg cells: CD4⁺CD127^{-/low}CD25⁺. Our parameters and markers for isolation of Tregs is therefore consistent with established procedures and the literature. Sorting of the CD4⁺CD127^{-/low}CD25⁺ population using these markers is the method of choice when isolating the Treg population, whether in culture or from animals, and to assess their suppression ability (Ellis et al., 2012). In the response to the reviewer's concern that FoxP3 was not included in the isolation scheme, it is because FoxP3 is a transcription factor and found in all activated T cells not just Tregs, detection of which involves cell fixation and permeabilization. Isolation of live Tregs never includes FoxP3 because it cannot be stained if downstream applications require preservation of intact and live cells. Because Reviewer #3 requested further explanation, we now note in the text that as shown by many groups (Hartigan-O'Connor et al., 2007; Liu et al., 2006; Seddiki et al., 2006; Yu et al., 2012), the majority of the cells that downregulate the IL-7 receptor (or CD127) are FoxP3⁺, in contrast to the majority of effector and memory T cells (Boettler et al., 2006; Fuller et al., 2005; Huster et al., 2004; Li et al., 2003) which re-express CD127 (Liu et al., 2006). The CD4⁺CD127^{-/low}CD25⁺ T cell subset expresses the highest level of FoxP3 in humans and the strongest immune suppressive capability (Yu et al., 2012), which is how we also identify, quantify and isolate these cells. In addition, both Treg suppressive function and FoxP3 levels inversely correlate with those of CD127 (Hartigan-O'Connor et al., 2007; Liu et al., 2006; Seddiki et al., 2006). Therefore, we use the very well-established combination of CD4, CD25 and CD127 that is crucial to obtain highly purified Treg cells that are distinguishable from conventional CD4⁺ T cells.

We would also like to point out again, as we mentioned previously in our Response to Reviewers and in our revised manuscript (pages 6 & 7), that upregulation CD25 (IL-2R alpha chain) is the expected outcome following IL-2 treatment. Moreover, it is critical and standard procedure to maintain T cell viability in culture by addition of IL-2, even in “untreated control cells”, as detailed in Methods. Therefore, CD25 upregulation is not an unspecific phenomenon that invalidates our well-established sorting strategy, but rather an expected and normal outcome of IL-2 treatment used in all T cell studies. Again, it is the combination of CD4, CD25 and CD127 markers and not only CD25 which defines a Treg cell.

Regarding JReviewer #3's suggestion that we characterize the non-Treg population (which is less than 40% and not 70% as indicated by the reviewer in the previous review), this experiment cannot add any relevant information to the manuscript. It is already well known that *in vitro*, naïve T cell stimulation (α CD3 and α CD28) along with IL-2 treatment in Treg-polarizing conditions gives rise to iTregs (CD4⁺CD127^{-low}CD25⁺), naïve CD4⁺ T cells (CD4⁺CD25⁻CD45RA⁻) and memory T cells (CD4⁺CD25⁻CD45RA⁺) (Ellis et al., 2012; Schmidt et al., 2016; Schmidt et al., 2018). There will not be other T helper subsets (e.g. Th1, Th2, Th17) present as we have not used the corresponding polarizing cytokines (Kaiko et al., 2008). This is now discussed in the revised manuscript (pages 8 & 9).

Regarding previous point 4, we thank Reviewer #3 for pointing out that we did not provide images of gating strategy. These are now provided by including representative flow cytometry plots for all remaining graphs for which they were not previously included. Representative flow cytometry image data are now provided for all plots, and have been added as new Supplementary Figure 3, Figure 4 new panel d, Figure 8 new panel d, and new Supplementary Figure 8.

Point 5, even if we follow the authors interpretation of the suppression data, that, it is the number of cells within each peak which denotes suppression and not the number of divided peaks (although in my opinion it is usually both!), if one looks at the cultures with TGF- β and TGF- β +RAD001, there doesn't seem to be a significant difference. In addition, the counts (y axis) are not similar in all the cohorts, hence the change in distribution of cells although may show suppressive potential, the data with RAD001 plus TGF- β is not biologically significant as compared to TGF- β alone.

We noted in our previous response to this issue by Reviewer #3, that the percent suppression is calculated by a well-established method used by virtually all investigators, and the division index is calculated by the FlowJo software that is universally accepted. Small differences in the y axis of the flow cytometry histograms are not significant and are due to slightly different numbers of cells recorded for each sample. Immunology papers do not normally explain the calculation of suppression in such depth for that reason, although in response to Reviewer #3, we now provide a very brief explanation in Methods.

We respectfully disagree with the reviewer regarding the significance of the suppression data. The difference between TGF- β and TGF- β +RAD001 is in fact biologically significant. We note for instance that the 8:1 ratio, which was found to be the most optimal condition, showed 2.6 times more suppression in the combined treatment than when treating with TGF- β alone, and was significant at $P < 0.01$.

Point 11- do the authors show viability test in 5d? or 6c? It would be important to perform viability test with Annexin V/ 7AAD analysis in order to determine apoptosis that occurs with RNA interference assays. Also, in Figure 8, I do not see any viability data.

Apparently, the reviewer missed these data which were provided in the previous versions of this manuscript. We note that there is no Fig. 5d - both Figures 5 and 6 show genome-wide data. It is therefore difficult to understand what lack of viability data the reviewer is referring to. The viability data in the prior revision of our manuscript is for HEK cells with DAP5 silencing (now moved to Supplemental Fig. 7c), and as viability was not changing, it is not relevant carry out further apoptosis/necrosis assays. For revised Figure 8 we

now provide data showing unchanged viability between non-silenced and DAP5 silenced matched cells at all conditions (new panel e).

Minor concern, expand on whether the error bars in the column graphs are a representation of SEM or SD not the MFI.

We thank the reviewer for noticing. This has now been added to the figure legend where it was missing.

REFERENCES

- Boettler, T., E. Panther, B. Bengsch, N. Nazarova, H.C. Spangenberg, H.E. Blum, and R. Thimme. 2006. Expression of the interleukin-7 receptor alpha chain (CD127) on virus-specific CD8+ T cells identifies functionally and phenotypically defined memory T cells during acute resolving hepatitis B virus infection. *J Virol* 80:3532-3540.
- Ellis, G.I., M.C. Reneer, A.C. Velez-Ortega, A. McCool, and F. Marti. 2012. Generation of induced regulatory T cells from primary human naive and memory T cells. *J Vis Exp*
- Fuller, M.J., D.A. Hildeman, S. Sabbaj, D.E. Gaddis, A.E. Tebo, L. Shang, P.A. Goepfert, and A.J. Zajac. 2005. Cutting edge: emergence of CD127high functionally competent memory T cells is compromised by high viral loads and inadequate T cell help. *J Immunol* 174:5926-5930.
- Hartigan-O'Connor, D.J., C. Poon, E. Sinclair, and J.M. McCune. 2007. Human CD4+ regulatory T cells express lower levels of the IL-7 receptor alpha chain (CD127), allowing consistent identification and sorting of live cells. *J Immunol Methods* 319:41-52.
- Huster, K.M., V. Busch, M. Schiemann, K. Linkemann, K.M. Kerksiek, H. Wagner, and D.H. Busch. 2004. Selective expression of IL-7 receptor on memory T cells identifies early CD40L-dependent generation of distinct CD8+ memory T cell subsets. *Proc Natl Acad Sci U S A* 101:5610-5615.
- Kaiko, G.E., J.C. Horvat, K.W. Beagley, and P.M. Hansbro. 2008. Immunological decision-making: how does the immune system decide to mount a helper T-cell response? *Immunology* 123:326-338.
- Li, J., G. Huston, and S.L. Swain. 2003. IL-7 promotes the transition of CD4 effectors to persistent memory cells. *J Exp Med* 198:1807-1815.
- Liu, W., A.L. Putnam, Z. Xu-Yu, G.L. Szot, M.R. Lee, S. Zhu, P.A. Gottlieb, P. Kapranov, T.R. Gingeras, B. Fazekas de St Groth, C. Clayberger, D.M. Soper, S.F. Ziegler, and J.A. Bluestone. 2006. CD127 expression inversely correlates with FoxP3 and suppressive function of human CD4+ T reg cells. *J Exp Med* 203:1701-1711.
- Schmidt, A., M. Eriksson, M.M. Shang, H. Weyd, and J. Tegner. 2016. Comparative Analysis of Protocols to Induce Human CD4+Foxp3+ Regulatory T Cells by Combinations of IL-2, TGF-beta, Retinoic Acid, Rapamycin and Butyrate. *PLoS One* 11:e0148474.
- Schmidt, A., F. Marabita, N.A. Kiani, C.C. Gross, H.J. Johansson, S. Elias, S. Rautio, M. Eriksson, S.J. Fernandes, G. Silberberg, U. Ullah, U. Bhatia, H. Lahdesmaki, J. Lehtio, D. Gomez-Cabrero, H. Wiendl, R. Lahesmaa, and J. Tegner. 2018. Time-resolved transcriptome and proteome landscape of human regulatory T cell (Treg) differentiation reveals novel regulators of FOXP3. *BMC Biol* 16:47.
- Seddiki, N., B. Santner-Nanan, J. Martinson, J. Zaunders, S. Sasson, A. Landay, M. Solomon, W. Selby, S.I. Alexander, R. Nanan, A. Kelleher, and B. Fazekas de St Groth. 2006. Expression of interleukin (IL)-2 and IL-7 receptors discriminates between human regulatory and activated T cells. *J Exp Med* 203:1693-1700.
- Yu, N., X. Li, W. Song, D. Li, D. Yu, X. Zeng, M. Li, X. Leng, and X. Li. 2012. CD4(+)CD25 (+)CD127 (low/-) T cells: a more specific Treg population in human peripheral blood. *Inflammation* 35:1773-1780.